# What Really Improves Mathematical Reasoning: Structured Reasoning Signals Beyond Pure Code

Yuze Zhao [1]   Junpeng Fang [2]   Lu Yu [2]   Zhenya Huang [1 3]   Kai Zhang [1]   Qing Cui [2]
Qi Liu [1 3]   JUN ZHOU [4]   Enhong Chen [1]

## Abstract

Code has become a standard component of modern foundation language model (LM) training, yet its role beyond programming remains unclear. We revisit the claim that code improves reasoning through controlled pretraining experiments on a 10T-token corpus with fine-grained domain separation. Our findings are threefold. First, when code is restricted to standalone executable programs and Code-NL data are controlled for, code substantially improves programming ability but does not act as a general reasoning enhancer; instead, it competes with knowledge-intensive tasks, especially complex mathematical reasoning. Second, the reasoning gains often attributed to code are better explained by cross-domain structured reasoning traces, such as code-text and math-text mixtures, rather than by executable code alone. Third, increasing the density of structured math-domain samples within a fixed math budget yields substantial gains on difficult mathematical reasoning while largely preserving programming performance, suggesting that cognitive scaffolds offer a targeted way to mitigate cross-domain trade-offs. Finally, routing analyses show that data-composition effects are reflected in expert-activation patterns, providing mechanism-level evidence for competitive and synergistic interactions across domains. Our results clarify which data characteristics transfer across capability dimensions and point to more precise data-centric optimization strategies.

[1]State Key Laboratory of Cognitive Intelligence, University of Science and Technology of China, Hefei, China [2]Individual Researcher [3]Institute of Artificial Intelligence, Hefei Comprehensive National Science Centerce [4]Zhejiang University, Hangzhou, China. Correspondence to: Kai Zhang <kkzhang08@ustc.edu.cn>.

*Proceedings of the 43rd International Conference on Machine Learning*, Seoul, South Korea. PMLR 306, 2026. Copyright 2026 by the author(s).

## 1. Introduction

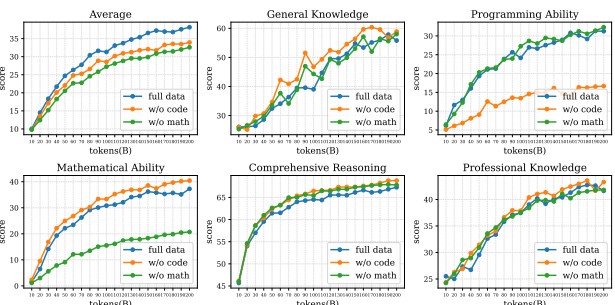

*Figure 1.* The impact of three data compositions on model performance across capability dimensions. Starting from the 10T-token corpus, we ablate either the code corpus (w/o code) or the math corpus (w/o math), and then evaluate the resulting models along five dimensions: general knowledge, coding ability, mathematical ability, comprehensive reasoning, and professional knowledge. The results suggest clear trade-offs under a fixed-token training budget: code data competes with knowledge-intensive tasks, especially mathematical reasoning, whereas math data competes with tasks requiring comprehensive cross-domain reasoning.

In general-purpose large language models (LLMs), code typically accounts for approximately 10%-30% of the pretraining corpus, making it a central component of modern training pipelines. Since the success of Codex (Chen et al., 2021), which demonstrated the effectiveness of fine-tuning language models on code, and InstructGPT (Ouyang et al., 2022), which showed that even a modest proportion of code data can substantially improve general-purpose performance, code has become a standard ingredient in LLM development. As LLMs have rapidly evolved, training on code has been associated with improvements in agentic behavior (Nakano et al., 2021; Kosinski, 2023; Chen et al., 2025), code reasoning (Chen et al., 2023; Gao et al., 2023; Zhao et al., 2025b), software-engineering (Yang et al., 2025; Sun et al., 2026), and tool use (Schick et al., 2023; Hong et al., 2024; Wu et al., 2024). Collectively, these advances underscore the influential role of code corpora in shaping contemporary LLM capabilities.

The benefits of incorporating code into training data are multifaceted. First, exposure to code substantially improves

performance on programming-related tasks (Zhao et al., 2024; 2025a). Second, compared to heterogeneous web text, code is typically more structured, less noisy, and of higher average quality, often resulting in lower training loss at convergence (Figure 7). Third, the explicit syntactic and structural constraints inherent in code can encourage models to produce outputs that are more logically organized and syntactically well-formed (OpenAI, 2024; Anthropic, 2024).

Beyond these well-established advantages for programming tasks, a growing line of work has investigated whether code data also benefits non-coding abilities, particularly reasoning. For example, Ma et al. (2024) studied the role of code in both pretraining and supervised instruction tuning, finding that mixed code-text pretraining improves reasoning performance across logical, scientific, analogical, and legal domains without evident negative transfer. Similarly, Aryabumi et al. (2025) examined the effects of code quality and corpus composition on natural-language reasoning, reporting that allocating roughly 25% of the training corpus to code yields the best overall performance. They further showed that high-quality synthetic code is more effective than large quantities of web-scraped code. Despite differences in experimental setup, these studies converge on a seemingly counterintuitive conclusion: incorporating code data into the training corpus enhances the reasoning abilities of general-purpose LLMs.

In this work, we revisit and critically reassess the claim that code enhances reasoning ability. We conduct large-scale, tightly controlled experiments by constructing a high-quality 10T-token corpus and pretraining Mixture-of-Experts (MoE) models of varying sizes from scratch. On this basis, we perform two systematic ablation studies: (i) disentangling algorithmic code, or pure code, from application-oriented code to isolate the contribution of the former, and (ii) ablating the mathematics portion of the corpus. Our results show that domain corpora do not contribute additively to model capabilities; instead, they exhibit pronounced competitive interactions. Specifically, pure code primarily competes with knowledge-intensive tasks, whereas math data mainly competes with comprehensive reasoning tasks. Incorporating pure code is associated with an overall performance drop of 14.38% on mathematical benchmarks. Conversely, including math data substantially improves performance on competitive programming tasks (up to 37.11%) but reduces code reasoning performance by as much as 17.30%.

We attribute the discrepancy between our findings and prior work in part to differences in corpus definition and granularity. Unlike previous studies that treat code as a monolithic category, our corpus construction enforces stricter standards for quality control and classification precision, explicitly distinguishing pure code from cross-domain code-text mixtures. In our setting, code is defined strictly as executable functions or program segments, excluding explanatory text, comments, and instructional descriptions. This distinction allows us to disentangle programming ability from cross-domain reasoning signals at the data level, thereby revealing their divergent effects on model capabilities. Under this formulation, we find that pure code alone does not directly enhance reasoning performance; instead, the gains reported in prior work may be partly attributable to mixed-reasoning data that integrate code, natural language, and mathematical structure.

Building on this insight, we further investigate whether a more targeted form of cognitive scaffolding can improve data efficiency. We identify a class of mathematically grounded samples that are highly structured and explicitly support multi-step reasoning. These samples improve complex reasoning performance without substantially degrading programming ability. By increasing their training proportion while holding the overall mathematical token budget constant, we observe significant gains on challenging mathematical reasoning benchmarks, while performance on code benchmarks remains largely unchanged, with a negligible drop of approximately 1%. These results suggest that structured reasoning signals can serve as a useful scaffold for high-difficulty reasoning tasks.

Finally, we analyze expert-routing patterns associated with cooperative and adversarial interactions among domain-specific data sources. By examining expert-routing distributions in the MoE model under different corpus configurations, together with hook-based analyses, we find that the proposed cognitive scaffolding improves complex reasoning while maintaining stable expert allocation. This evidence is consistent with the role of structured reasoning data as a cross-domain signal that enhances reasoning performance without destabilizing other competencies.

## 2. Related Work

### 2.1. Data Influence on LLM Reasoning

Data constitute the fundamental source of model capabilities (Schaeffer et al., 2023; Razeghi et al., 2022). Understanding how data from different domains, particularly code and mathematics, interact, cooperate, or compete within LLMs has emerged as a central theme in recent research. Ma et al. (2024) investigated the impact of introducing code corpora at different training stages and concluded that pretraining on mixed code and text substantially enhances reasoning abilities, including logical, scientific, analogical, and legal reasoning, with little evidence of negative transfer. Similarly, Aryabumi et al. (2025) demonstrated that incorporating high-quality synthetic code during both pretraining and annealing improves reasoning performance, while cautioning that an excessive proportion of code data, more

than $3/4$ of the corpus, severely compromises performance on knowledge-intensive tasks. Collectively, these studies support the view that code can play an important role in strengthening reasoning and generalization.

In contrast, our work revisits this consensus from a critical perspective. By conducting experiments with a similar overarching design but finer-grained corpus definitions, we arrive at a different conclusion. Building on this divergence, we further identify the data factor that more directly accounts for improved reasoning ability, which we term cognitive scaffolding.

## 2.2. Data Selection and Data Mixing

Data selection and data mixing represent two complementary strategies for optimizing training corpora. These approaches operate at different levels of granularity: instance-level optimization and domain-level optimization, respectively. Data selection focuses on identifying the most valuable pretraining instances, thereby accelerating convergence and improving downstream performance (Albalak et al., 2024). Depending on when they are applied, selection methods can be categorized as offline or online. Offline methods filter corpora before training, using techniques such as data pruning (Marion et al., 2023; Tirumala et al., 2023) and data programming (Ratner et al., 2016; Zhou et al., 2025). Online methods, in contrast, dynamically adjust sampling strategies during training. For example, Gu et al. (2025) employed Pontryagin's Maximum Principle to prioritize high-quality data that provides the steepest descent in gradient-based optimization, while Jiang et al. (2025) estimated domain-level loss during training to preferentially sample more promising data, thereby improving learning efficiency.

Data mixing addresses the problem of determining optimal sampling ratios across domains to maximize overall model performance. DoReMi leverages group distributionally robust optimization to train a small proxy model without requiring prior knowledge of downstream tasks. The proxy model then generates domain weights that are used to resample large-scale corpora for LLM training (Xie et al., 2023). Another line of work, such as REGMIX, trains multiple proxy models under different data-mixing strategies and uses regression to predict model performance across ratios, thereby inferring an optimal mixture strategy (Liu et al., 2025b).

Building on these foundations, this paper integrates data selection and data mixing into a unified perspective. Adjusting mixture ratios enables a systematic exploration of cooperative and competitive dynamics across domains; the insights gained from this exploration then inform the data selection stage, allowing us to identify and prioritize instances that exhibit cross-domain synergy.

## 3. Experimental Setup

### 3.1. Model Architecture

For our experiments, we employ MoE models of varying scales. In this architecture, the standard feed-forward network (FFN) is replaced by a collection of $N$ experts (Fedus et al., 2022; Lepikhin et al., 2021; Jiang et al., 2024), each implemented as a compact modular FFN unit. This design improves both computational efficiency and expert specialization. The MoE model dynamically routes each token to a subset of experts through a router $R$, defined as follows:

$$g_t = \text{Softmax}(R(o_t)),$$
$$p_t = \sum_i g_{t,i} E_i(o_t) \quad \text{s.t.} \quad g_{t,i} \in \text{TopK}(g_t), \quad (1)$$

where $o_t \in \mathbb{R}^d$ is the $d$-dimensional output of multi-head attention, $E_i$ represents the $i$-th expert, $g \in \mathbb{R}^N$ denotes the gating vector, and $p_t$ is the output representation of the $t$-th token.

To further improve training efficiency and scalability, we adopt a fine-grained expert strategy on top of the standard MoE (Dai et al., 2024; Liu et al., 2024). While keeping the total number of model parameters fixed, we increase the number of experts and reduce the intermediate dimensionality of each expert. This configuration promotes stronger expert specialization. To preserve a general-purpose pathway despite the reduced capacity of individual experts, we introduce an additional shared expert $E_s$ trained on all tokens. The resulting output is:

$$p'_t = p_t + E_s(o_t). \quad (2)$$

An essential component of the MoE architecture is the routing module. In our design, we adopt a dropless routing strategy together with load-balancing loss and router z-loss to improve training efficiency and prevent uneven token allocation across experts. To further mitigate instability during the early stages of pretraining, we introduce a mechanism termed stochastic routing warmup. This method injects controlled randomness into the routing process, thereby alleviating expert overloading and reducing the risk of expert collapse caused by severe routing imbalance early in training stage.

Formally, let $s_t \in \mathbb{R}^N$ denote the routing logits for an input token representation $h_t \in \mathbb{R}^{d'}$, computed by a linear projection layer. During the warmup phase, when the current step $t_c$ is smaller than the warmup step $t_w$, we interpolate between the learned logits and synthetic random logits. The final routing logits $s'_t$ are given by:

$$s_t = W^{\text{T}} h_t + b,$$
$$s'_t = \alpha s_t + (1 - \alpha)(\mu_s + \sigma_s \cdot \epsilon), \quad \epsilon \sim \mathcal{N}(0, 1), \quad (3)$$

where $W \in \mathbb{R}^{d' \times N}$ is a projection matrix, $\alpha = \min(\frac{t_c}{t_w}, 1.0)$ is the warmup coefficient, and $\mu_s$ and $\sigma_s$ are the running mean and standard deviation of $s_t$, respectively.

## 3.2. Data Preparation and Domain Division

We construct a comprehensive pretraining corpus of approximately 10T tokens, spanning seven major domains: Web, Code, Code-NL, Math, Wikipedia, Books, and Multilingual. To ensure model quality, we apply standardized procedures for data acquisition, curation, and admission control. The pipeline is designed to preserve dataset quality, balance, and utility across domains.

### 3.2.1. DATA ACQUISITION AND CURATION PIPELINE

The data pipeline comprises three stages: collection, curation, and admission control. Together, these stages produce high-quality, domain-balanced training inputs.

In the collection phase, raw data are sourced from diverse public repositories, including Common Crawl (Patel, 2020), GitHub (Hellendoorn & Sawant, 2022), arXiv (Clement et al., 2019), Project Gutenberg (Stroube, 2003), Wikipedia dumps (Wikimedia Foundation, 2026), and multilingual web archives (Braud et al., 2024). Code is collected from open-source repositories across major programming languages. Mathematical content is drawn from arXiv and educational platforms, and is augmented with internally generated synthetic datasets covering core concepts and proof patterns.

During curation, each data type is subjected to targeted filtering to remove noise, structural errors, and low-signal content. Code files are validated for syntax, length, duplication, and functional density; math data are checked for LaTeX correctness, expression well-formedness, and explanatory coherence. Processing parameters and language-specific rules are detailed in Appendix A.4.

In the admission control stage, segments are scored using a fine-grained framework of more than 300 metrics across 10 dimensions, including coherence and factual consistency. Domain-specific scoring criteria, such as algorithmic richness for code and derivation depth for math, yield tiered labels (high, medium, and low), which determine sampling weights during pretraining (Raffel et al., 2020). This enables prioritized learning from high-signal content while maintaining cross-domain balance.

### 3.2.2. DOMAIN CATEGORIZATION

The corpus is organized into seven well-defined, semantically coherent domains, each serving distinct cognitive and linguistic functions in model learning. **Web**: general-purpose natural language from diverse online sources, providing breadth and real-world linguistic variation. **Code**: standalone executable functions, scripts, and program segments from open-source repositories, emphasizing syntactic precision and algorithmic logic while excluding surrounding explanations, instructional descriptions, and comments during curation. **Code-NL**: mixed code-and-language data that occupy an intermediate position between source code and natural language. In form, such data exhibit clear structural characteristics resembling code; in content and intended use, they are not primarily designed for conventional programming tasks. Instead, they support application-oriented problems in areas such as mathematics, finance, scientific research, and numerical computation, where code snippets are often interleaved with problem descriptions, explanations, or derivations. **Math**: formal mathematical expressions, proofs, derivations, and problem-solution pairs that foster symbolic reasoning. **Wikipedia**: structured encyclopedic knowledge with cross-referenced facts, supporting factual grounding and conceptual understanding. **Books**: long-form narrative and expository texts that promote discourse coherence and deep semantic comprehension. **Multilingual**: high-quality parallel and monolingual texts across major world languages, enabling cross-lingual transfer.

We distinguish between Code and Code-NL primarily according to data source and rule-based classification criteria. Code data are mainly collected from GitHub repositories, where code density exceeds 60% after comment and boilerplate removal and the content is explicitly intended to solve programming-related problems. In contrast, Code-NL data are primarily retrieved from web-based or mixed-format sources. This taxonomy enforces strict boundaries to prevent overlap and ambiguity, particularly between mixed-source data, such as notebooks, Q&A pages, Markdown documents, HTML/CSS fragments, and standalone programming artifacts. For instance, code embedded within explanatory text is categorized as Code-NL, whereas standalone code repositories are classified as Code. In our pure-code ablations, Code-NL is retained; therefore, the intervention isolates the marginal contribution of standalone algorithmic code rather than removing mixed reasoning traces that contain code-like syntax.

### 3.2.3. DATA ORGANIZATION AND MIXING STRATEGY

Data organization follows a two-phase strategy: *quality-tiered stratification* and *balanced mixture scheduling*. After cleaning and scoring, each domain is divided into high-, medium-, and low-quality strata. Only data above a predefined threshold enter the final training mixture.

During training, we employ a dynamic sampling policy that balances token-level contributions while prioritizing high-value domains such as Math and Code. Guided by ablation studies on data contributions, this mixture ensures sufficient exposure to reasoning-intensive content without overwhelming the general-language distribution. We validate the strat-

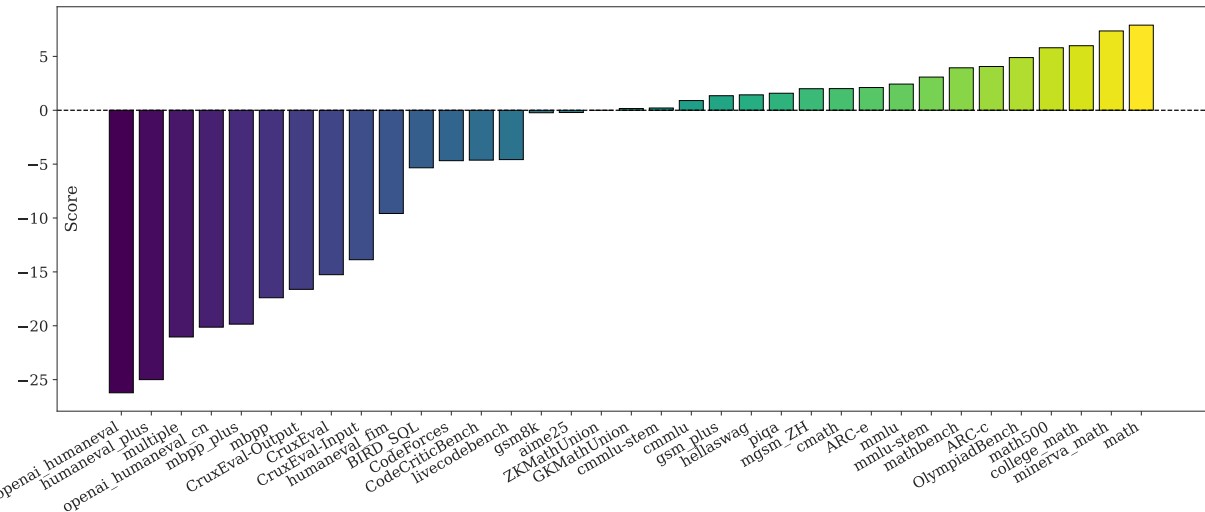

*Figure 2.* Code data exhibits a competitive relationship with knowledge-intensive tasks. When code is ablated from the full corpus, performance declines substantially across all programming benchmarks, as expected. Beyond programming, code data also competes with comprehensive reasoning tasks such as PIQA and HellaSwag. For mathematical reasoning, the impact is more task-dependent: code data significantly hinders performance on complex benchmarks such as MATH and OlympiadBench, while its effect on simpler problems such as GSM8K remains comparatively limited.

egy through controlled experiments on small-scale models before large-scale deployment, ensuring stable convergence and balanced capability development.

## 4. Experimental Observations

We construct a 10T-token training corpus categorized into seven domains: Web, Code, Code-NL, Math, Wikipedia, Books, and Multilingual. We begin with a central question: *How do code and mathematics corpora influence overall model performance?* To investigate this question, we perform controlled interventions by ablating specific data subsets from the full corpus and measuring the resulting changes in downstream performance. Keeping the total number of training tokens fixed, we separately remove the pure-code corpus and the math corpus. Data from the remaining domains are upsampled proportionally to fill the vacated training budget. This setup measures fixed-budget substitution effects: the observed differences compare models that allocate part of the training budget to the target domain against models that redistribute the same budget across the remaining domains, rather than simply changing the total amount of training data. We evaluate the resulting models across five capability dimensions: general knowledge, programming ability, mathematical ability, comprehensive reasoning, and professional knowledge. The benchmark-to-dimension mapping and aggregate score definition are provided in Appendix A.3.

Contrary to interpretations that code broadly improves reasoning, our results reveal asymmetric trade-offs under a

fixed training budget. We use *negative coupling* to describe cases in which adding data from one domain improves in-domain performance but reduces performance in another capability dimension under a fixed training budget. This phenomenon is analogous to interference in transfer learning, where optimization signals from one domain can impede learning in another. In our experiments, code data is associated with lower performance on knowledge-intensive tasks, while math data is associated with reduced gains in comprehensive reasoning. Such negative coupling points to a fixed-budget trade-off in multi-domain training, where gains in one capability dimension may come at the expense of another.

### 4.1. Code Data Competes with Mathematical and Knowledge-intensive Tasks

We compare MoE models trained on the full corpus with models trained after ablating code data; the results are shown in Figure 2. The experiments reveal a clear intra-domain effect: ablating code markedly diminishes the programming ability of LLMs, as expected. Beyond programming, the full-data model with code underperforms the `w/o code` model on several non-programming benchmarks under the fixed-token mixture. For instance, on comprehensive reasoning benchmarks such as PIQA and HellaSwag, including code reduces performance by 2.11% and 2.39%.

For mathematical reasoning, the competitive effect of code is more pronounced and strongly dependent on task difficulty. Under this fixed-budget comparison, the full-data model with code scores 14.38% lower on average across

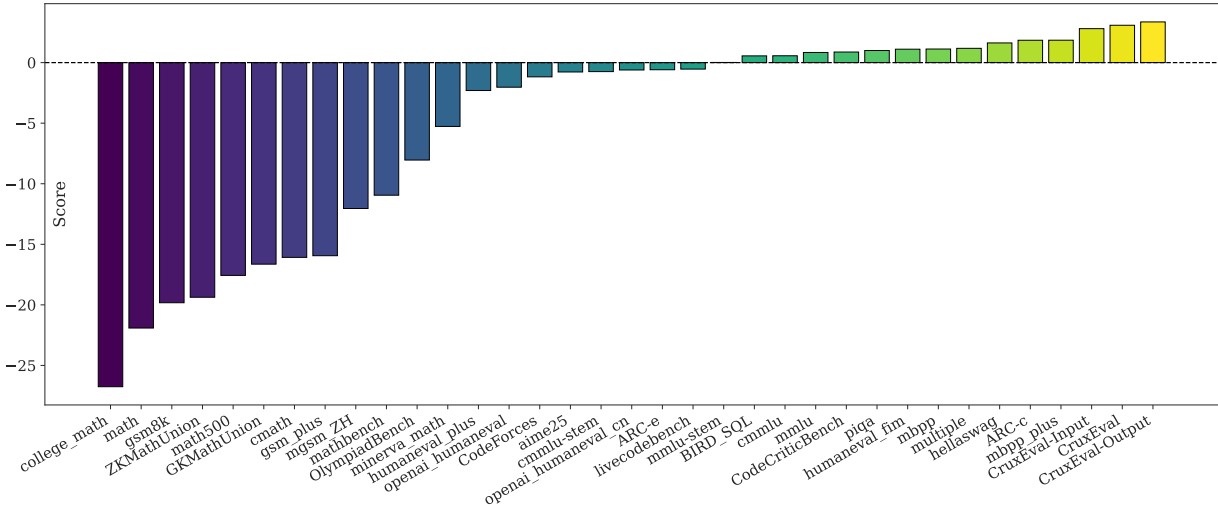

*Figure 3.* Math data exhibits a competitive effect on comprehensive reasoning tasks. As with code, ablating math data causes a pronounced decline on mathematical benchmarks. Unlike code, however, math data exerts limited competitive influence on programming ability. Its competitive effect is more apparent in comprehensive reasoning tasks, including code reasoning benchmarks such as CruxEval and MBPP and commonsense reasoning benchmarks such as HellaSwag.

mathematical benchmarks than the `w/o code` model, although the impact varies across benchmarks. For less structurally demanding benchmarks, such as GSM8K (+0.4%) and CMath (-3%), the effect is small or mixed. For more complex benchmarks, the detrimental impact is substantial: Minerva-Math (-71.53%), OlympiadBench (-47.16%), MATH (-22.64%), CollegeMath (-12.48%), and MathBench (-10.23%) all exhibit performance degradation.

## 4.2. Math Data Competes with Comprehensive Reasoning

As with code data, ablating the math corpus weakens model performance within its own domain. However, in contrast to the strong competitive effect that code exerts on mathematics, the influence of math data on programming is less pronounced and shows divergent trends across programming tasks. On competitive programming benchmarks, incorporating math data markedly improves performance, for example on CodeForces (+37.11%) and LiveCodeBench (+11.26%). This suggests that mathematical data can provide useful algorithmic and symbolic problem-solving signals for competitive programming tasks. By contrast, on code reasoning tasks that require hybrid reasoning patterns, math data is associated with lower performance, for example on CruxEval (-17.30%) and MBPP (-6.12%). A similar negative trend is observed on commonsense reasoning benchmarks, such as HellaSwag (-2.94%), where the inclusion of math data coincides with measurable degradation. We observe the same qualitative pattern on dense models at 1B and 5B scales and on MoE variants with 16 and 32 experts, suggesting that the trade-off is less likely to be specific to a

single model size or routing configuration. Detailed results are provided in Appendix A.8.

We next examine why these results differ from prior findings. At first glance, our conclusions may seem to contradict prior studies, but the two views are not necessarily incompatible. During data preparation, we explicitly distinguish pure code data from Code-NL data, and we retain Code-NL in the code ablation experiments. Through this control, we better separate the marginal effect of pure executable code from that of cross-domain structured data.

It is important to note that, although these cross-domain data are collected from programming-related websites, they are not inherently code-centric. For instance, Aryabumi et al. (2025) treated markup languages such as Markdown, CSS, and HTML as code for training purposes. However, these data primarily originate from web text and are presented in structured formats. As a result, they contain substantial non-code knowledge, which may help explain why prior work observed improvements across non-code domains when broader code or markup-like data were grouped under code.

This distinction is important for interpreting the sign of the ablation effects. If Code and Code-NL were removed together, the resulting performance change would conflate two different sources of signal: executable programming structure and mixed-format reasoning traces embedded in natural language. In our setting, Code-NL is retained in both the `full data` and `w/o code` configurations, so it functions as a control for structured but non-programming content. The code ablation therefore provides a fixed-budget estimate of the marginal contribution of standalone exe-

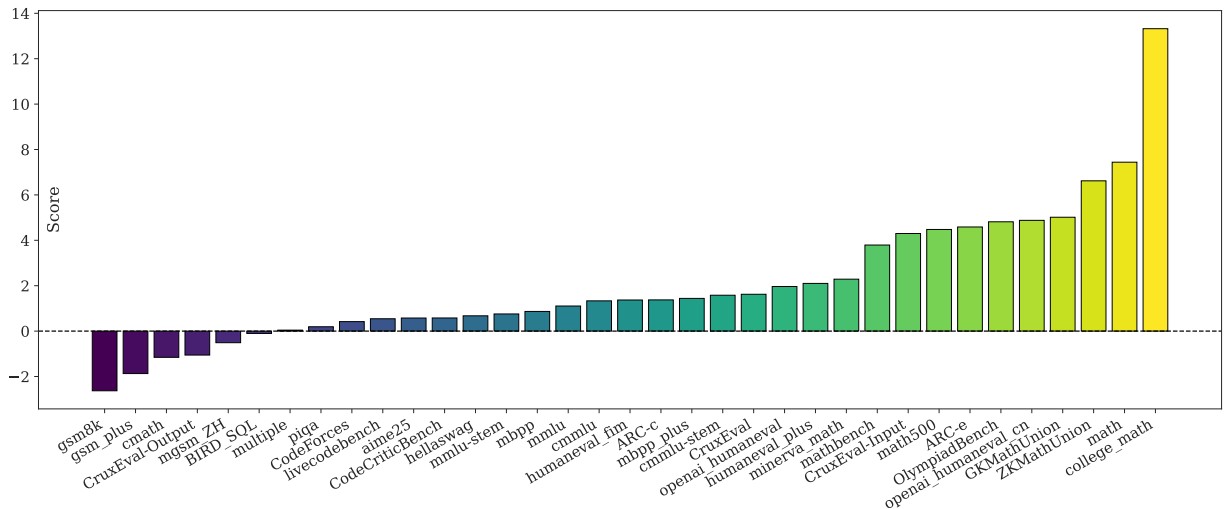

*Figure 4.* Incorporating structured reasoning data affects tasks differently. For challenging datasets such as College Math and MATH, cognitive scaffolds improve model performance. By contrast, for tasks that can often be solved without explicit structured reasoning, such as GSM8K and CMath, adding such data introduces competition and may hinder performance.

cutable code, while holding mixed code-language reasoning traces in the training mixture.

Based on analysis, we conclude that although data from different domains often exhibit negative coupling, useful cross-domain intersections may also exist. These observations motivate the next question: whether structured cross-domain intersections can be selected more precisely to improve reasoning while limiting degradation in other capabilities.

## 5. Cognitive Scaffolding for Complex Mathematical Reasoning

Building on these observations, we aim to identify data subsets that improve reasoning performance under controlled training conditions, which we refer to as cognitive scaffolds. We use this term in an operational sense rather than as a universal formal category: in this paper, a cognitive scaffold is a math-domain sample selected by a structural identifier because it exposes intermediate reasoning structure. Such samples typically contain explicit subgoals, symbolic manipulations, hierarchical derivation steps, case decompositions, or verification procedures, making the reasoning trajectory more visible and traceable than in ordinary natural-language solutions (Liu et al., 2025a).

Formally, let $\mathcal{D}_{\mathrm{math}}$ denote the math-domain corpus, $f_\theta(x)$ denote the structural-classifier score for sample $x$, and $\tau$ denote the threshold calibrated on the validation set. The cognitive-scaffolding subset used in our experiments is defined as:

$$\mathcal{D}_{\mathrm{cog}} = \{x \in \mathcal{D}_{\mathrm{math}} \mid f_\theta(x) \geq \tau\}. \qquad (4)$$

Thus, a sample is included because it is selected by the

structural classifier within the math corpus, not because it matches a manually specified rule over symbol density, indentation, length, or step count.

These data primarily originate from the math domain and exhibit a high degree of structural organization. They improve model reasoning performance, particularly on complex reasoning tasks, without competing strongly with code data. Cognitive scaffolds expose intermediate symbolic and procedural structure, which may help models learn patterns useful for multi-step reasoning, causal analysis, and cross-domain comprehension.

To identify cognitive scaffolds, we need an approach that balances efficiency and effectiveness, especially at trillion-token scale. Concretely, we search for math-domain data with strong structural signatures and treat this structure as a key signal for cognitive scaffolding. To this end, we train a lightweight FastText (Bojanowski et al., 2017) classifier to capture structural features in text. The classifier is trained with 200,000 code samples as positive instances and 200,000 randomly sampled non-code instances as negative examples, enabling it to detect structured patterns efficiently. These code-derived positives are used only as a scalable proxy for external organization, not as a source of programming semantics for the final math data. We then apply the trained FastText model to the math corpus to identify structured reasoning samples and use them as cognitive scaffolds.

The filtering procedure is calibrated to favor precision over recall, since false positives could contaminate the scaffold subset with ordinary mathematical prose. We train the classifier on approximately 400k samples and evaluate it on 188,678 validation samples. The resulting classifier achieves

*Table 1.* Post-hoc structural statistics of the original MATH corpus and the selected cognitive-scaffolding subset. These statistics are diagnostic only; they were not used as explicit hand-crafted filtering rules during scaffold extraction.

| Feature | MATH | Cognitive scaffolds |
|---|---|---|
| Symbol density | 0.0363 | 0.0518 |
| Avg. derivation steps | 3.3720 | 4.4124 |
| Indentation ratio | 0.0006 | 0.5446 |
| Text length | 531 | 2821 |

an accuracy of 0.9696, positive-class precision of 0.9998, and positive-class recall of 0.9665. We also conduct a contamination audit on the selected scaffold data to check for residual code segments and unintended formatting artifacts, helping ensure that the filter captures structural reasoning patterns rather than mathematical-content leakage. Detailed selection criteria are provided in Appendix A.9.

Table 1 shows that the selected data contain richer structural organization than the original MATH corpus. Our extraction pipeline does not explicitly filter by symbol density, derivation-step count, indentation ratio, or length. These properties emerge as post-hoc characteristics of the samples selected by FastText, suggesting that the lightweight classifier recovers useful structural signals without relying on manually specified thresholds for these features.

We observe that cognitive scaffolds exhibit several hallmarks of "structured reasoning". Formally, such data are highly organized, often characterized by explicit symbolic systems, hierarchical derivation procedures, and rigorous logical chains. They therefore expose transparent reasoning trajectories and traceable intermediate steps. In content, these data are closely aligned with mathematical reasoning, covering paradigms such as formula derivation, proof construction, equation solving, and function modeling. These tasks demand both precise local symbolic manipulation and globally coherent cross-step dependencies.

We then incorporate these structured reasoning data into training while keeping the overall proportion of math-domain data unchanged. Concretely, cognitive scaffolds replace part of the original math-domain data rather than adding extra tokens or increasing the total math proportion. After this within-domain replacement, the resulting math data are sampled through the same training pipeline and sampling strategy used in the other settings; cognitive scaffolds are not treated as a separate domain with a special schedule. Because we do not conduct a systematic sweep over replacement ratios, we do not claim that more replacement is always better. Our claim is narrower: under a fixed math budget, increasing the density of structured reasoning samples improves difficult mathematical reasoning.

The experimental results are presented in Figure 4. Cognitive scaffolds yield particularly notable improvements on complex reasoning tasks: on average, overall mathematical reasoning performance increases by 17.56%. This benefit comes with a trade-off. On relatively simple tasks, such as GSM8K and CMath, structured reasoning can disrupt cases that could otherwise be solved directly through natural language, leading to performance drops of 6.29% and 2.00%, respectively. Nevertheless, these losses are outweighed by substantial gains on complex tasks. For instance, we observe significant improvements on College Math (+30.05%), MATH (+23.17%), OlympiadBench (+47.78%), and MathBench (+14.51%). These results support the importance of structured reasoning signals in complex mathematical problem-solving, while also highlighting their competitive interaction with unstructured natural-language reasoning.

Beyond mathematics, cognitive scaffolds have potential for cross-domain transfer. Their embedded patterns of logical deduction and abstract structure may benefit non-mathematical scenarios such as program verification, experimental design, and causal modeling. Thus, structured reasoning data may serve not only mathematical reasoning but also complex tasks across domains.

## 6. Data Composition Shapes Expert Routing Patterns

Finally, we examine how cooperative and adversarial interactions among data sources are reflected in the MoE model's internal routing behavior. To this end, we analyze expert-routing distributions under varying data configurations during autoregressive generation across domains. Specifically, we examine activated experts in Math (GSM8K and MATH500), Code (HumanEval and Live-CodeBench), and question answering (MMLU), and quantify distributional discrepancies using Jensen-Shannon (JS) divergence (Menéndez et al., 1997). For each domain, Figure 5 reports expert-level routing-probability deviations relative to the full-data model for the 20 experts with the largest absolute deviations, together with pairwise JS divergence computed over the complete expert-routing distribution. The corresponding analysis over all 64 experts is reported in Appendix A.6.

At the expert level, the Code domain exhibits a pronounced redistribution under the `w/o code` configuration: several of the most affected experts show deviations in opposite directions relative to the full-data model, and the corresponding JS divergence is large. This indicates that code data do not merely improve downstream programming performance; they also help maintain a distinct expert-allocation pattern for code-like inputs. In the Math domain, ablating mathematical data also changes the routing pattern substantially, while the response is not confined to the matched

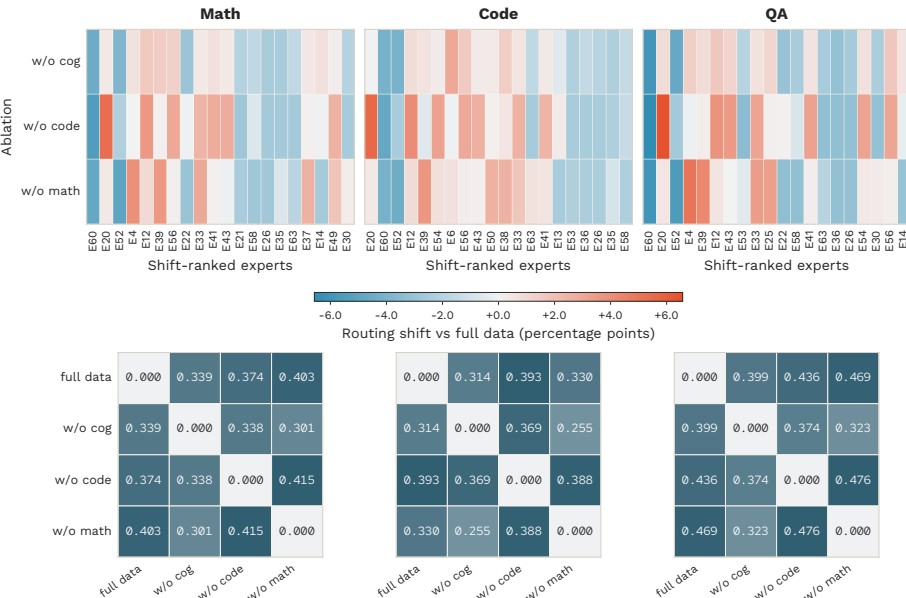

*Figure 5.* MoE expert-routing probability deviations and JS divergence in the Math, Code, and QA domains under different data configurations. The upper row shows the 20 experts with the largest absolute deviations relative to the full-data model within each domain; the lower row reports pairwise JS divergence between complete expert-routing distributions.

ablation alone. The effect of code-related ablation on part of the Math-domain routing distribution is consistent with the downstream evidence of coupling between structured code signals and mathematical reasoning data.

In contrast, the scaffold-related ablation, `w/o cog`, induces comparatively smaller and more diffuse expert-level deviations across domains. This suggests that cognitive scaffolds contribute mainly by improving transferable reasoning structure rather than by forcing a broad reallocation of domain-specific experts. In other words, cognitive scaffolds enhance complex reasoning while largely preserving the stability of expert-activation distributions, suggesting that they can function as cross-domain stabilizing signals within LLMs.

## 7. Conclusion and Future Work

In this work, we reexamined the counterintuitive claim that "code data enhances reasoning ability" through controlled experiments and fine-grained domain separation. We established a standardized pipeline for data collection and cleaning, yielding a 10T-token corpus constructed under strict data admission policies. Focusing on two key domains, Code and Math, we designed corpus configurations with high intra-domain cohesion and low inter-domain coupling, and systematically evaluated their effects through ablation studies. Our results show that code corpora exert strong competitive effects on knowledge-intensive tasks, particularly complex mathematical reasoning, while math corpora interfere more strongly with comprehensive reasoning. Based on these observations, we identify cognitive scaffolds that

support complex reasoning. Incorporating such data yields improvements in complex mathematical reasoning under conditions of low cross-domain competition.

Finally, by analyzing changes in expert-activation distributions across corpus configurations, we provide routing-level evidence for how data composition is reflected in the internal dynamics of LLMs, giving rise to competitive or synergistic effects across domains. Taken together, our findings highlight a promising path for data-centric model optimization: under fixed budgets, appropriately incorporating cross-domain structured reasoning data can mitigate cross-domain trade-offs, preserving performance in one domain while enhancing it in another.

## Impact Statement

This work aims to improve the understanding of how pre-training data composition affects LM capabilities. The study is empirical and does not introduce a deployed system. Its main societal relevance lies in supporting more transparent and efficient data-centric model development.

## Acknowledgment

This research was supported by the National Natural Science Foundation of China (Grants No. 62525606, 62477044, 62406303), the Key Technologies R&D Program of Anhui Province (No. 202423k09020039), the Young Elite Scientists Sponsorship Program by CAST (No. 2024QNRC001), and the Fundamental Research Funds for the Central Universities (No. WK2150110038).

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

# A. Appendix

## A.1. Ethics Statement

This work adheres to the ICML Code of Ethics. No human-subject research or animal experimentation was involved. All datasets were sourced in compliance with relevant usage guidelines, with care taken to avoid privacy violations. No personally identifiable information was used, and no experiments were conducted that would raise privacy or security concerns. We also took care to avoid introducing biases or discriminatory outcomes in the research process and are committed to maintaining transparency and integrity throughout the study.

## A.2. Reproducibility Statement

We have made every effort to ensure that the results presented in this paper are reproducible. All code and datasets have been made publicly available in an anonymous repository to facilitate replication and verification. The experimental setup, including training steps, model configurations, and hardware details, is described in detail in the paper. We also provide the complete training protocol in Appendix A.5 to support verification of the reported experiments.

## A.3. Benchmark-to-Dimension Mapping

We group downstream benchmarks into five capability dimensions according to the primary ability targeted by each benchmark. For a capability dimension $c$ with benchmark set $\mathcal{B}_c$, the aggregate score is computed as the arithmetic mean of the benchmark scores:

$$S_c = \frac{1}{|\mathcal{B}_c|} \sum_{b \in \mathcal{B}_c} s_b, \tag{5}$$

where $s_b$ denotes the normalized score on benchmark $b$. When a benchmark is unavailable for a particular model family, the aggregate is computed over the available benchmarks in the same dimension.

*Table 2.* Benchmark-to-dimension mapping used for capability aggregation.

| Capability dimension | Benchmarks |
|---|---|
| General knowledge | ARC-c, ARC-e |
| Programming ability | HumanEval, HumanEval-FIM, MBPP, MBPP-Plus, CruxEval, CodeCriticBench, Code-Forces, BIRD-SQL, Multipl-E, LiveCodeBench |
| Mathematical ability | GSM8K, CMath, Minerva-Math, AIME25, MATH/MATH500, OlympiadBench, College-Math, MathBench, GKMathUnion, ZKMathUnion |
| Comprehensive reasoning | PIQA, HellaSwag |
| Professional knowledge | CMMLU, CMMLU-STEM, MMLU, MMLU-STEM |

## A.4. Data Processing Details

This appendix provides details on the data curation pipeline described in Section 3.2.1, including code filtering rules, mathematical content validation procedures, and synthetic data generation protocols. These components are designed to ensure a high signal-to-noise ratio and structural integrity across domains.

### A.4.1. CODE DATA CLEANING RULES

To maintain high-quality code inputs, we apply a multi-stage, language-specific filtering framework. The following table summarizes the core cleaning rules applied to source files collected from GitHub and other public repositories.

Key definitions:

- **Executable code**: lines containing function bodies, control flow statements, or expressions; excludes imports, comments, type annotations, and empty lines.

- **Normalized AST**: abstract syntax trees with identifiers anonymized and constants replaced to detect semantic duplicates.

*Table 3.* Code cleaning rules by programming language.

| Filter Type | Python | JavaScript | Java | C++ | General |
|---|---|---|---|---|---|
| Long line filter | > 200 chars per line | > 150 chars per line | > 150 chars per line | > 150 chars per line | Apply soft wrap detection |
| File length filter | < 10 or > 10k tokens | < 10 or > 8k tokens | < 15 or > 9k tokens | < 15 or > 9k tokens | Exclude stubs and artifacts |
| Syntax validation | AST parsing via `libcst` | ESTree AST (via `espree`) | Javalang parser | Tree-sitter | Reject invalid syntax |
| Format check | PEP8 compliance check | ESLint basic rules | Check braces/indentation | GCC pre-parse | Remove malformed layout |
| Low-quality language filter | Filter docstrings-only | Filter HTML-in-JS (e.g., JSX only) | Filter interface-only files | Filter header-only if trivial | Remove boilerplate-heavy files |
| Language-specific filters | No `.pyc`, `__pycache__` | No minified (`.min.js`) | No auto-generated (Lombok, etc.) | No generated bindings | Block known bad patterns |
| Signal density threshold | < 30% executable code (after comment removal) → discard | Same rule applied across all languages | | | |
| Deduplication level | Function-level (via normalized AST) and repo-level (SimHash) | Retain canonical implementation | | | |

- **Signal density**: ratio of executable code lines to total file size after preprocessing.

All filtering steps are executed sequentially. Files failing any critical rule (syntax, format, signal density) are discarded. Surviving files are further deduplicated before inclusion in the training corpus.

### A.4.2. MATHEMATICAL CONTENT VALIDATION

For mathematical data sourced from arXiv and internal generators, we enforce strict structural and semantic checks to preserve correctness and pedagogical value.

**Validation Pipeline**

The processing pipeline for math content follows these stages:

1. **Source normalization**: Convert PDFs and HTML to structured text using `GROBID` and custom LaTeX extractors.

2. **LaTeX syntax check**: Validate all math environments ($, $$, `equation`, `align`) using `LaTeXML` and regex-based grammar rules.

3. **Expression well-formedness**: Detect unbalanced parentheses, mismatched delimiters, undefined commands, and incomplete integrals/sums.

4. **Consistency check**: Ensure equation labels are referenced, theorem-environment nesting is correct, and variables are consistently used.

5. **Noise detection**: Flag segments with excessive placeholders, repeated derivations, or formula-to-text ratio exceeding 80%.

6. **Final filtering**: Remove entries failing two or more checks above.

### A.5. Training Details

We pretrain a 20-layer autoregressive MoE language model with a hidden size of 2048 and 16 attention heads. Each MoE layer contains 16 experts with an intermediate size of 2048, and the feed-forward network (FFN) hidden size in dense layers is set to 5120. The MoE router selects the top-2 experts per token and employs a sigmoid score function with a router bias update rate of 0.001.

Training is conducted using a pipeline-parallel and tensor-parallel setup with a global batch size of 2048 and a micro-batch size of 4. Sequence parallelism and gradient accumulation fusion are enabled to optimize memory usage. The model is trained with bfloat16 and FP8 mixed precision.

Optimization is performed with AdamW ($\beta_1 = 0.9, \beta_2 = 0.95$, weight decay=0.1) and a constant learning rate of $5 \times 10^{-5}$, with 2,000 warmup iterations. Training uses a block LM objective with masked softmax fusion, SwiGLU activation, and unidirectional attention. The embedding weights and output projection are untied, and no norm head is used. Training runs for 24,000 iterations. Model checkpoints are saved every 1,200 iterations.

*Table 4.* Detailed results across MoE and dense model settings.

### MATHEMATICAL ABILITY

| Model | gsm8k | cmath | minerva_math | aime25 | math500 | OlympiadBench | GKMathUnion | ZKMathUnion | Overall |
|---|---|---|---|---|---|---|---|---|---|
| full data (32e) | 54.59 | 65.39 | 9.93 | 2.92↑ | 26.40 | 10.22↑ | 41.59↑ | 59.35 | 36.20 |
| w/o code (32e) | 56.48↑ | 70.13↑ | 13.60↑ | 1.67 | 33.60↑ | 9.33 | 36.51 | 65.73↑ | 38.52↑ |
| w/o math (32e) | 25.25 | 47.81 | 4.04 | 0.00 | 7.00 | 1.19 | 19.37 | 38.87 | 17.71 |
| full data (16e) | 46.40 | 59.93↑ | 9.93 | 0.42 | 24.00 | 14.22↑ | 31.59 | 56.53 | 31.60 |
| w/o code (16e) | 49.73↑ | 59.29 | 13.24↑ | 1.04↑ | 27.00↑ | 10.81 | 31.90↑ | 57.27↑ | 32.91↑ |
| w/o math (16e) | 22.29 | 42.90 | 5.15 | 0.42 | 8.80 | 2.37 | 14.29 | 36.20 | 15.99 |
| full data (5B) | 64.59↑ | 73.04 | 18.01↑ | - | 38.00 | 15.56↑ | 23.65 | 64.54↑ | 45.13 |
| w/o code (5B) | 63.08 | 73.68↑ | 14.34 | - | 45.40↑ | 12.91 | 33.97↑ | 63.95 | 46.56↑ |
| w/o math (5B) | 44.28 | 57.19 | 6.25 | - | 11.60 | 2.67 | 20.48 | 45.99 | 26.71 |
| full data (1B) | 26.99 | 45.17 | 7.35↑ | - | 22.80 | 4.15↑ | 20.00 | 42.58 | 20.08 |
| w/o code (1B) | 29.57↑ | 48.27↑ | 6.99 | - | 25.60↑ | 0.59 | 21.60↑ | 44.96↑ | 25.11↑ |
| w/o math (1B) | 10.99 | 30.42 | 2.57 | - | 3.00 | 2.07 | 6.40 | 23.44 | 9.66 |

### PROGRAMMING ABILITY

| Model | humaneval | humaneval_fim | mbpp | mbpp_plus | CruxEval | CodeCriticBench | CodeForces | BIRD_SQL | Multipl-E | LiveCodeBench | Overall |
|---|---|---|---|---|---|---|---|---|---|---|---|
| full data (32e) | 46.95↑ | 59.05 | 33.00 | 36.77 | 24.81↑ | 50.72 | 5.62↑ | 4.01 | 28.12 | 4.90 | 26.94↑ |
| w/o code (32e) | 17.68 | 46.47 | 14.20 | 17.46 | 10.56 | 53.49 | 1.17 | 0.49 | 7.20 | 0.65 | 14.25 |
| w/o math (32e) | 38.41 | 60.31↑ | 34.80↑ | 39.68↑ | 21.12 | 54.33↑ | 3.93 | 5.18↑ | 29.00↑ | 5.56↑ | 24.25 |
| full data (16e) | 57.89 | 37.80↑ | 39.15↑ | 34.15↑ | 19.06↑ | 55.35↑ | 1.17↑ | 3.29 | 25.62↑ | 6.05↑ | 26.94↑ |
| w/o code (16e) | 46.76 | 13.41 | 14.29 | 12.80 | 9.06 | 52.09 | 1.17↑ | 0.29 | 5.66 | 0.49 | 14.25 |
| w/o math (16e) | 58.86↑ | 37.20 | 34.13 | 31.10 | 14.25 | 52.70 | 1.17↑ | 4.73↑ | 25.04 | 5.39 | 24.25 |
| full data (5B) | 48.78 | 63.02↑ | 42.68 | 45.50 | 35.94↑ | 56.53↑ | 1.17 | 9.81↑ | 28.78↑ | 8.50 | 35.38↑ |
| w/o code (5B) | 25.00 | 52.76 | 20.12 | 22.75 | 28.06 | 53.19 | 3.93 | 2.71 | 9.08 | 4.58 | 23.14 |
| w/o math (5B) | 49.39↑ | 62.15 | 45.12↑ | 45.77↑ | 34.94 | 52.47 | 5.85↑ | 6.94 | 17.97 | 9.31↑ | 34.10 |
| full data (1B) | 24.39↑ | 51.11 | 19.20↑ | 21.34↑ | 13.38↑ | 52.81 | 1.17 | 1.79↑ | 14.92 | 0.65 | 18.71↑ |
| w/o code (1B) | 2.44 | 39.79 | 4.00 | 1.83 | 5.44 | 53.26 | 1.17↑ | 0.13 | 2.39 | 0.00 | 9.09 |
| w/o math (1B) | 9.15 | 51.89↑ | 18.00 | 7.93 | 11.94 | 53.77↑ | 1.17↑ | 1.56 | 15.78↑ | 1.31↑ | 15.99 |

### GENERAL KNOWLEDGE

| Model | ARC-c | ARC-e | Overall |
|---|---|---|---|
| full data (32e) | 41.02 | 64.37 | 52.29 |
| w/o code (32e) | 48.81↑ | 66.49↑ | 53.48 |
| w/o math (32e) | 43.73 | 62.79 | 53.89↑ |
| full data (16e) | 43.73 | 60.85↑ | 52.29 |
| w/o code (16e) | 46.10 | 60.85↑ | 53.48 |
| w/o math (16e) | 47.46↑ | 60.32 | 53.89↑ |
| full data (5B) | 58.31 | 74.96 | 66.64 |
| w/o code (5B) | 61.02↑ | 78.84↑ | 69.93↑ |
| w/o math (5B) | 60.00 | 77.78 | 68.89 |
| full data (1B) | 28.47 | 34.04 | 31.26 |
| w/o code (1B) | 36.27↑ | 47.44↑ | 41.86↑ |
| w/o math (1B) | 34.24 | 43.74 | 38.99 |

### COMPREHENSIVE REASONING

| Model | piqa | hellaswag | Overall |
|---|---|---|---|
| full data (32e) | 50.21 | 73.34 | 64.83 |
| w/o code (32e) | 54.33↑ | 74.97↑ | 66.34↑ |
| w/o math (32e) | 20.23 | 74.05 | 65.55 |
| full data (16e) | 73.45 | 56.20 | 64.83 |
| w/o code (16e) | 74.81↑ | 57.86↑ | 66.34↑ |
| w/o math (16e) | 73.50 | 57.60 | 65.55 |
| full data (5B) | 76.12↑ | 61.60 | 68.86 |
| w/o code (5B) | 76.12↑ | 62.82 | 69.47 |
| w/o math (5B) | 75.95 | 63.04↑ | 69.50↑ |
| full data (1B) | 70.51 | 47.41 | 58.96 |
| w/o code (1B) | 71.60 | 48.92 | 60.26 |
| w/o math (1B) | 72.25↑ | 48.96↑ | 60.61↑ |

### PROFESSIONAL KNOWLEDGE

| Model | cmmlu | cmmlu-stem | mmlu | mmlu-stem | Overall |
|---|---|---|---|---|---|
| full data (32e) | 35.90 | 39.18 | 43.22 | 44.93 | 39.06 |
| w/o code (32e) | 37.36↑ | 40.42↑ | 42.72 | 45.43↑ | 39.16 |
| w/o math (32e) | 36.54 | 36.44 | 43.48↑ | 43.48 | 39.80↑ |
| full data (16e) | 38.77 | 35.01 | 43.22 | 39.23 | 39.06 |
| w/o code (16e) | 40.57 | 36.10↑ | 42.72 | 37.26 | 39.16 |
| w/o math (16e) | 41.18↑ | 35.09 | 44.38↑ | 38.54 | 39.80↑ |
| full data (5B) | 47.51 | 41.16 | 52.03 | 45.86 | 46.64 |
| w/o code (5B) | 48.98↑ | 42.64↑ | 53.84↑ | 48.59↑ | 48.51↑ |
| w/o math (5B) | 46.71 | 38.69 | 52.31 | 45.31 | 45.76 |
| full data (1B) | 32.13 | 28.82 | 33.85 | 29.61 | 31.10 |
| w/o code (1B) | 35.97↑ | 32.09↑ | 39.53↑ | 35.32↑ | 35.73↑ |
| w/o math (1B) | 31.13 | 27.95 | 33.92 | 31.06 | 31.02 |

## A.6. Complete Expert-Routing Analysis

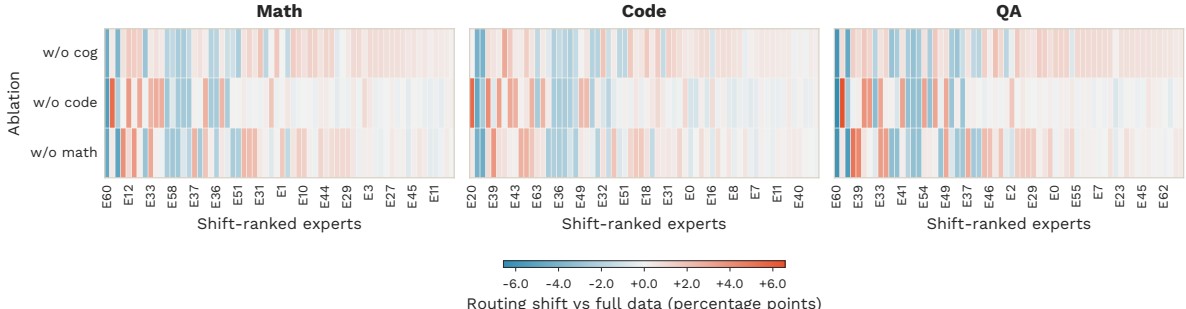

*Figure 6.* Complete 64-expert analysis of routing-probability deviations in the Math, Code, and QA domains. Experts are sorted within each domain by their maximum absolute deviation relative to the full-data model. This figure complements the top-20 expert-level summary in Figure 5.

## A.7. Detailed Experimental Results

Detailed experimental data are provided in Table 4. The results indicate a clear competitive relationship between the code and math corpora. In contrast, competitive effects in general knowledge, comprehensive reasoning, and professional

knowledge are smaller and less consistent.

## A.8. Conclusions Hold Across Architectures and Model Sizes

To validate the robustness and consistency of our conclusions, we trained two additional groups of model variants under settings matched as closely as possible to the original setup. First, we replaced the MoE architecture with dense models at 1B and 5B scales while maintaining consistent training data and computational budgets. Second, we adjusted the total parameter count in the MoE architecture by reducing the number of experts from 64 to 32 (32e) or 16 (16e).

As summarized in Table 4, the results are consistent with those reported in the main text. This indicates that the qualitative trends are robust across the tested architectures and model sizes. Within these experiments, model performance is primarily affected by parameter count, data volume, and computational budget.

At a finer granularity, we also observe a cross-domain budget trade-off: under a fixed total budget, allocating more resources to the code domain reduces the available budget for mathematical data, leading to performance degradation in the latter.

## A.9. Detailed Selection Criteria of Cognitive Scaffolds

The objective of cognitive scaffolding is to identify segments that exhibit "structured reasoning patterns" in large-scale math corpora. Operationally, let $\mathcal{D}_{\mathrm{math}}$ denote the math-domain corpus, $f_\theta(x)$ denote the FastText structural classifier score for sample $x$, and $\tau$ denote the threshold calibrated on the validation set. The cognitive-scaffolding subset used in our experiments is defined as:

$$\mathcal{D}_{\mathrm{cog}} = \{x \in \mathcal{D}_{\mathrm{math}} \mid f_\theta(x) \geq \tau\}. \tag{6}$$

Thus, a sample is included because it is selected by the structural classifier within the math corpus, not because it matches a manually specified rule over symbol density, indentation, length, or step count.

To construct $f_\theta$, we developed a binary classifier that distinguishes structured from unstructured text. The training set was constructed as follows:

### A.9.1. POSITIVE SAMPLE CONSTRUCTION

Positive samples were sourced from our rigorously curated high-quality code dataset, which contains approximately 200k instances. These samples were selected to exclude mathematical content while retaining explicit structured logic and formal expressions, such as indentation rules, identifier patterns, operator sequences, and function structures. Tree-sitter was employed to ensure that all code samples were parseable.

All positive samples originated from publicly available, reproducible code sources (GitHub repositories, online judge datasets, and internally compliant open-source mirrors), with deduplication and near-duplicate removal applied.

### A.9.2. NEGATIVE SAMPLE CONSTRUCTION

Negative samples were drawn from a broad web-text corpus. To reduce code-like structures, we applied lenient but effective regex-based heuristics to exclude "code-structured" elements, including:

- Removing texts containing high-frequency code symbol patterns (e.g., =, ::, int);
- Filtering out segments with multi-line indentation, consecutive {}, or grouped operators;
- Excluding texts containing programming language keywords, code file extensions, or Markdown formatting.

Only natural-language-dominated, syntactically loose texts were retained. This ensured broad coverage of negative samples while reducing the risk of introducing misclassified "weakly structured" texts.

All samples were cleaned according to Appendix A.4 before being admitted to the training pipeline.

### A.9.3. TRAINING AND THRESHOLD SETTING PRINCIPLES

We collected approximately 400k training samples and 188,678 validation samples. A lightweight structure identifier was trained using FastText. The classification threshold was determined by two principles: (1) maximizing the F1 score on the

validation set; and (2) prioritizing precision for the positive, structured class to avoid misclassifying ordinary mathematical text as "structured reasoning".

### A.9.4. CONTAMINATION AUDIT

All positive samples were code-based and had no overlap with mathematical data. Negative samples were sourced from web-based natural language and were distinct from both mathematical training corpora and code-related data.

The classifier was used only to detect "structured reasoning patterns" and was not trained on mathematical content. A sampled audit of the filtered cognitive-scaffolding data confirmed the absence of residual code segments or unintended formatting contamination. Thus, the filter is designed to avoid information leakage regarding mathematical content and to preserve fairness in subsequent mathematical evaluations.

### A.9.5. CLASSIFIER PERFORMANCE AND CALIBRATION

The FastText classifier was evaluated on the validation set, yielding an accuracy of 0.9696, positive-class precision of 0.9998, and positive-class recall of 0.9665. In constructing cognitive scaffolds, we prioritized precision over recall because false positives could contaminate the structured reasoning data and undermine their effectiveness for mathematical reasoning. To improve structural consistency and interpretability, we adopted a conservative classification threshold, enabling the classifier to achieve high precision on the validation set while maintaining sufficient recall.

### A.9.6. MISCLASSIFICATION ANALYSIS

**False Positives.** False positives primarily originate from web texts with "pseudo-structured" formats, such as heavily bulleted lists, pseudocode layouts, or Markdown tables. Although rare, these cases reflect the model's sensitivity to structural cues.

**False Negatives.** False negatives mostly involve weakly structured code snippets embedded in long texts, such as single-line function prototypes or unindented expressions. Their impact is limited because our scaffolding construction emphasizes high precision: false negatives reduce the number of candidate scaffolds but do not introduce erroneous structures.

### A.9.7. REPRESENTATIVE DATA FORMAT

Below we provide an illustrative cognitive-scaffold-style sample. The example illustrates the type of structured mathematical data selected by the scaffold filter: the solution is organized around explicit subgoals, intermediate reductions, cross-step dependencies, and a lightweight verification block.

```
Problem:
Find all integers n such that
    n^2 + n + 1 divides n^5 + n^2 + 1.

Step 1: Setup
    Let d = n^2 + n + 1. We aim to evaluate
        n^5 + n^2 + 1 (mod d).

Step 2: Key modular reduction
    Observe that
        n^3 - 1 = (n - 1)(n^2 + n + 1),
    so
        n^3 = 1 (mod d).
    Therefore,
        n^5 = n^2 * n^3 = n^2 (mod d),
    and hence
        n^5 + n^2 + 1 = 2n^2 + 1 (mod d).

Step 3: Reduce the divisibility condition
    Since d = n^2 + n + 1, we have
        2n^2 + 1 - 2d = -2n - 1.
    Thus the original condition implies
        d divides 2n + 1.

Step 4: Bound the possible integer solutions
```

```
For n >= 2,
    d = n^2 + n + 1 > 2n + 1,
so divisibility is impossible.
For n <= -3, write m = -n >= 3. Then
    d = m^2 - m + 1 > 2m - 1 = |2n + 1|,
so divisibility is also impossible.
It remains to check n in {-2, -1, 0, 1}.

Verification:
    Direct substitution gives valid solutions for
        n = -2, -1, 0, 1.

Conclusion:
    The complete solution set is {-2, -1, 0, 1}.
```

### A.10. Training Process Records

We record the training-loss trajectories to inspect how each data configuration affects optimization across domains. Figure 7 reports the baseline curves for the aggregate, Web, Math, and Code domains. Figures 8 and 9 compare the corresponding ablated configurations against the full-data baseline, making the domain-specific effects of removing code, math, or cognitive-scaffolding data explicit.

Overall, the records provide a direct optimization-side view of the main experimental settings. The aggregate curve summarizes the broad convergence pattern, while the domain-specific panels separate Web, Math, and Code behavior so that downstream performance changes can be interpreted together with the corresponding training dynamics.

### A.11. Detailed Downstream Performance

We further report detailed downstream performance over training tokens for each benchmark family. Figure 10 groups the curves by capability dimension: the average score, comprehensive reasoning and general knowledge, mathematical ability, professional knowledge, and programming ability. Each panel compares the full-data model with the ablated settings, revealing whether a data removal primarily changes broad average performance or a specific capability dimension.

The average and dimension-level panels summarize broad capability trends, while the individual benchmark curves expose task-specific sensitivity to data composition. This view complements the aggregate results in the main text by showing where the full-data setting, the `w/o code` setting, and the `w/o math` setting diverge during training.

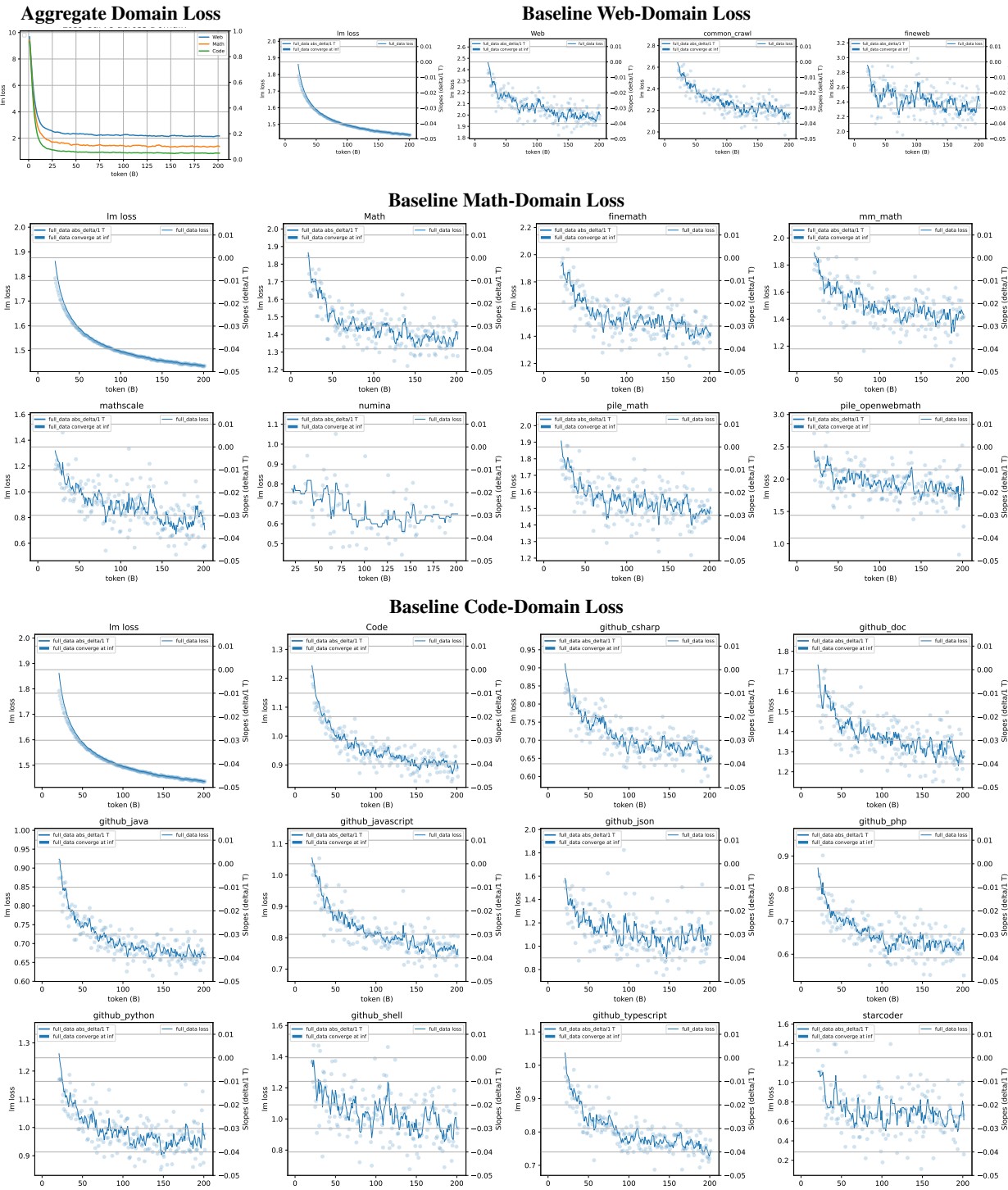

*Figure 7.* Baseline training-loss trajectories across aggregate, Web, Math, and Code domains.

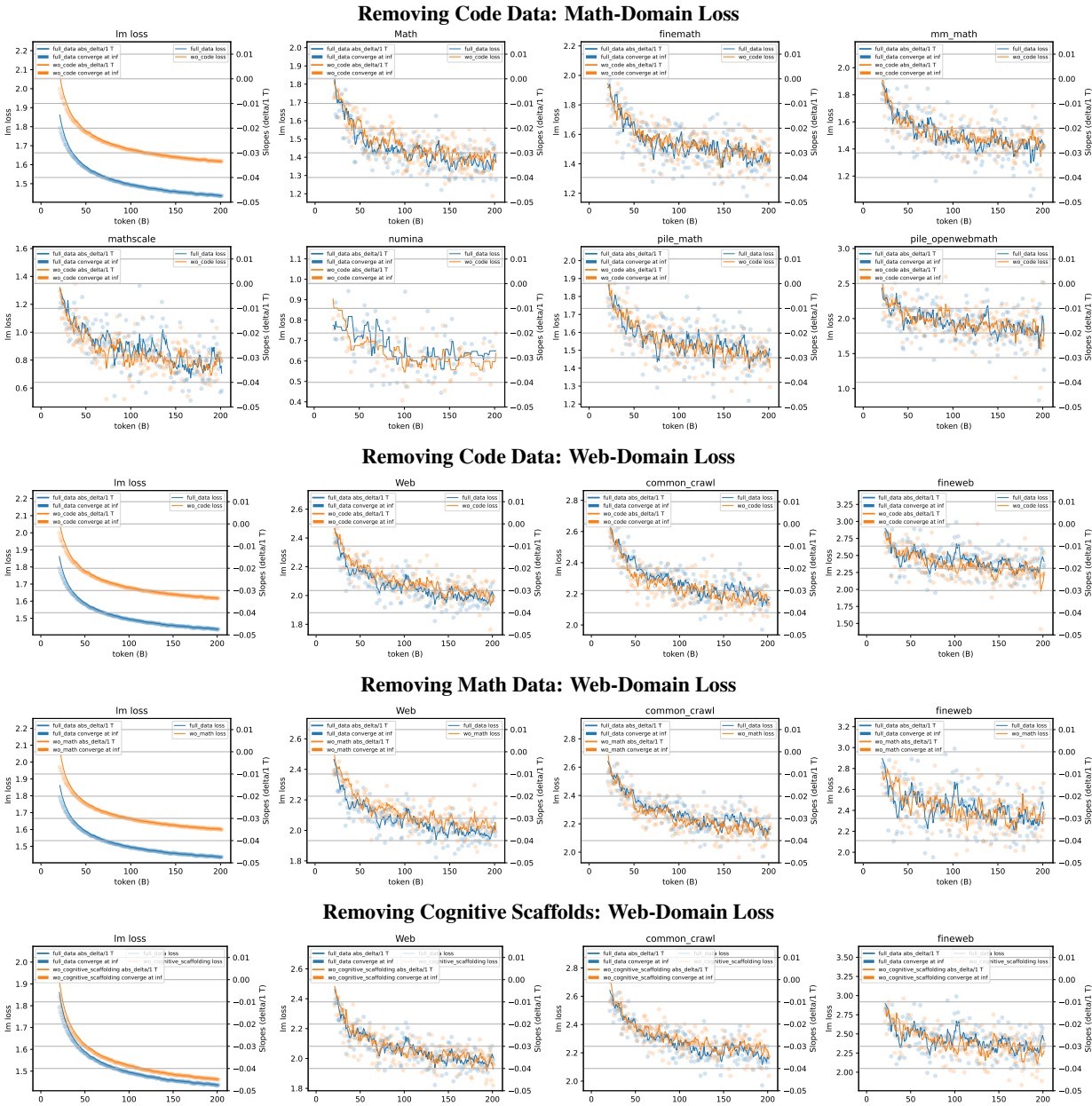

*Figure 8.* Math- and Web-domain training-loss comparisons for ablated data configurations.

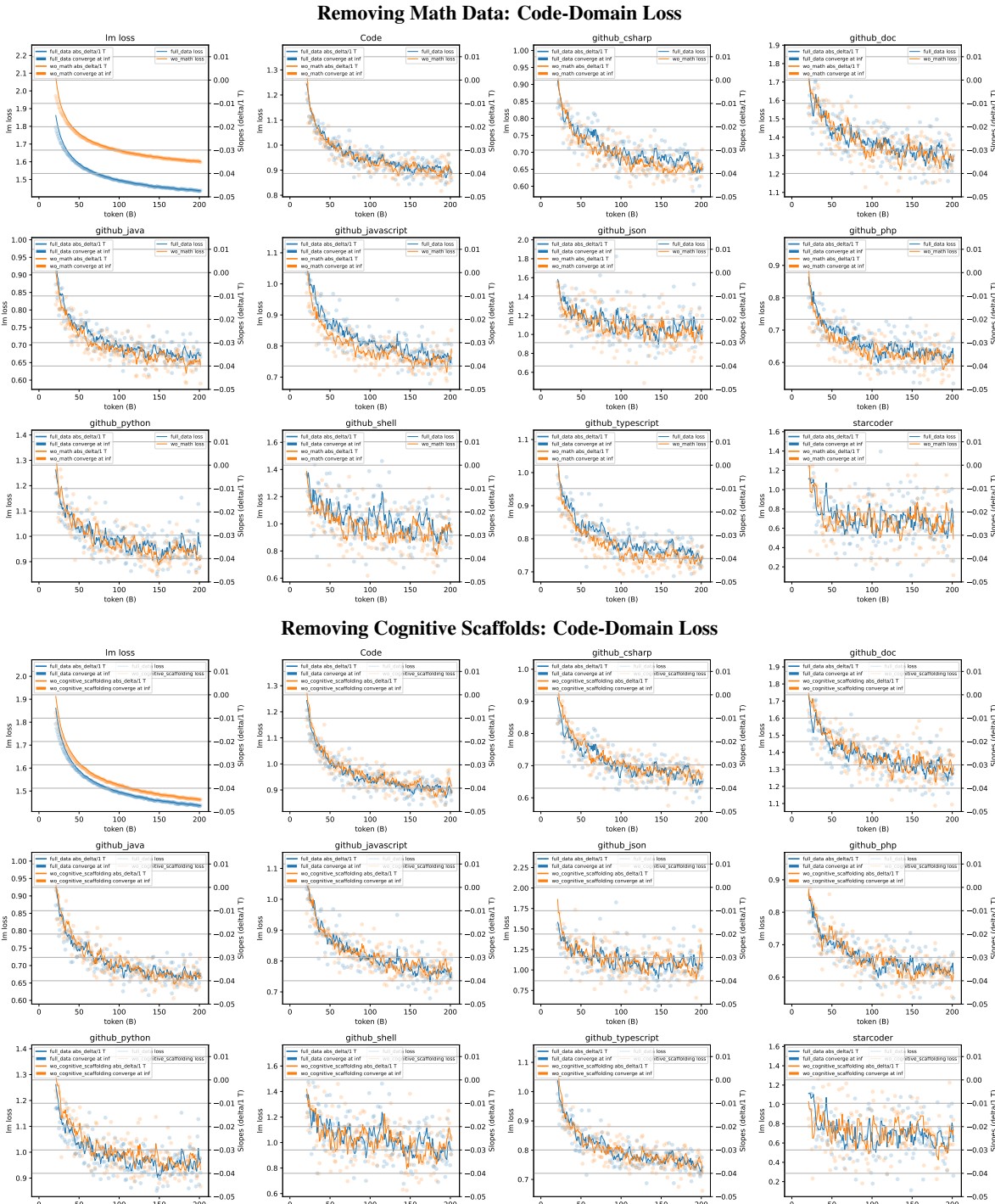

*Figure 9.* Code-domain training-loss comparisons for ablated data configurations.

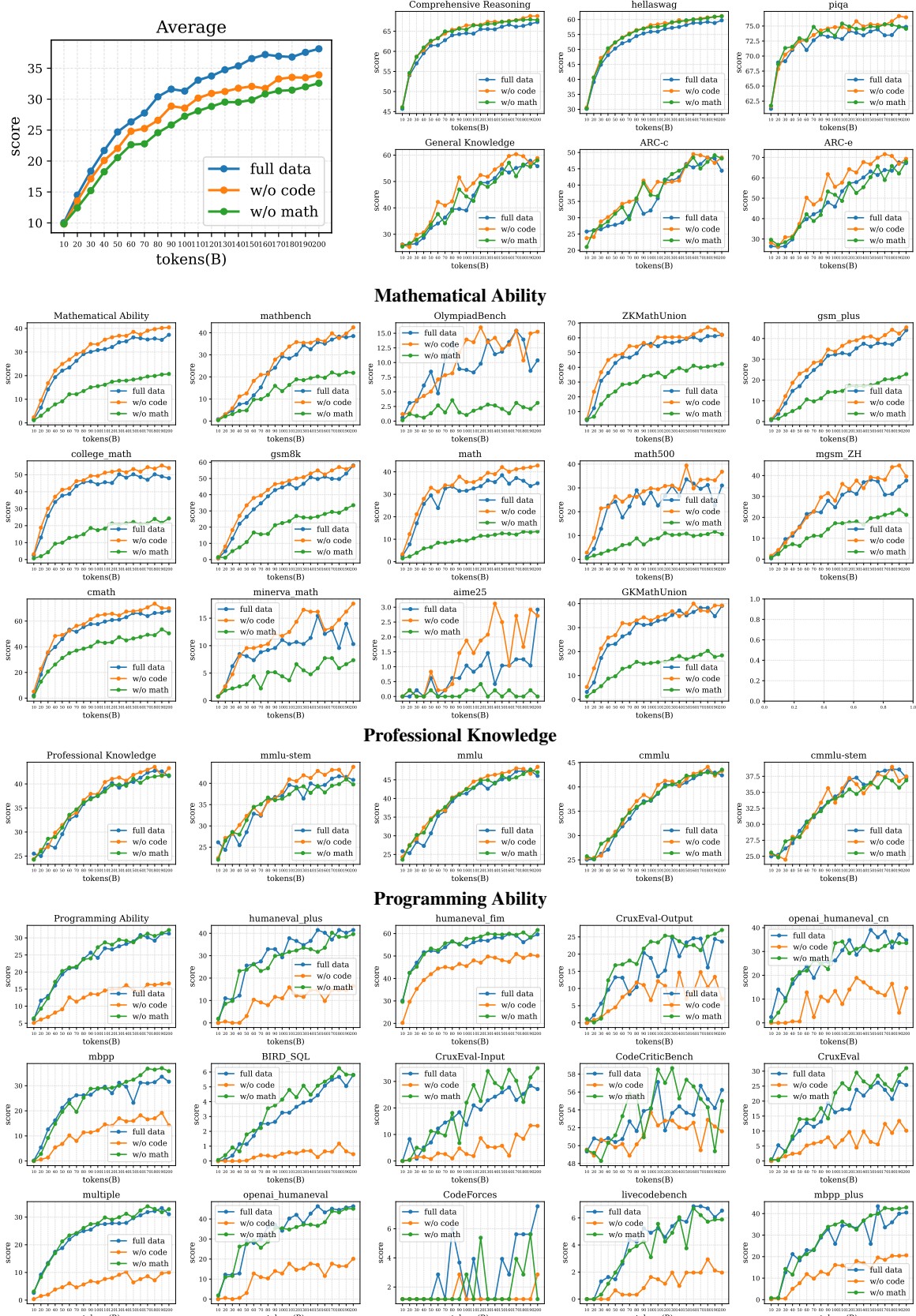

*Figure 10.* Evaluation results of different data configurations across all benchmarks.

