# OpenReview forum: "What Really Improves Mathematical Reasoning: Structured Reasoning Signals Beyond Pure Code"
_ICML.cc/2026/Conference — ICML 2026 regular_

### Official Review · Reviewer_EDiN · 2026-02-22

**Soundness:** 2
**Presentation:** 3
**Significance:** 2
**Originality:** 2
**Overall Recommendation:** 4
**Confidence:** 3

**Summary:**

This paper critically re-evaluates the claim that code data enhances reasoning abilities in LLMs through large-scale pretraining experiments. The authors refine the taxonomy of code by distinguishing between "pure code" and "code-NL" (code-natural language mixtures), finding that pure code degrades mathematical performance and that the true driver of improved reasoning is not pure code but structured mixed data. They introduce the concept of "cognitive scaffolds." The experimental scale is substantial, but the methodology suffers from fundamental flaws.

**Compliance With Llm Reviewing Policy:**

Affirmed.

**Final Justification:**

The authors’ rebuttal, particularly the controlled experiment isolating capacity competition from resource displacement, fully resolves my initial methodological concerns, which supports acceptance.

**Key Questions For Authors:**

- How can you demonstrate that "code-NL" improves reasoning because of its "code+text" hybrid format, rather than because it inherently contains textualized reasoning processes such as problem descriptions and logical derivations? If the latter holds, does your finding merely replicate the common knowledge that "training on text containing reasoning processes improves reasoning"?

- The selection criterion for "cognitive scaffolds" is "make math data statistically resemble code," yet the paper concludes that pure code is harmful to math. Is there an inherent contradiction here? If "resembling code" is beneficial, why is "being code" harmful?

- Your ablation studies were conducted under a fixed total budget. How do you distinguish whether the observed performance changes are due to capacity competition between domains or simply a resource allocation issue of differential data exposure? Are there controlled experiments that could support the former interpretation?

**Limitations:**

- The study cannot rule out the confounding factor that "semantic content inherently containing reasoning processes" drives the observed effects.

- The definition and selection of "cognitive scaffolds" are heavily dependent on the statistical features of code data, raising questions about the concept's independence and generalizability.

- The fixed-budget ablation design conflates "capacity competition" with "resource competition," undermining the causal conclusions.

**Strengths And Weaknesses:**

**Strengths:**

- Substantial Experimental Scale: Pretraining MoE models from scratch on a 10T-token corpus with systematic ablations demonstrates significant effort.

- Attempt at Fine-Grained Data Taxonomy: Distinguishing between "pure code" and "code-NL" offers a potentially valuable analytical perspective.

**Weaknesses:**

- Fundamental Confusion in Causal Inference: The authors attribute the performance difference between "pure code" and "code-NL" to the format feature of "whether it contains natural language." However, the two differ fundamentally in data source and semantic intent. "Pure code" from GitHub is a product of solving programming problems; "code-NL" from web tutorials/Q&A inherently contains textualized reasoning processes like problem descriptions and logical deductions. Observing that "code-NL" improves reasoning may simply be because it constitutes high-quality reasoning data itself, rather than because it is a "code+text" hybrid format. The authors fail to rule out this more direct explanation, invalidating their causal claim.

- Severe Definitional Ambiguity and Circularity in the Core Concept of "Cognitive Scaffolds": The appendix reveals that a FastText classifier was trained to identify "structured reasoning patterns," with positive training samples originating from code data. This implies the selection criterion is effectively "make math data look like code." This creates two problems: ① If "looking like code" is beneficial, why is "pure code" itself (according to the paper's conclusion) harmful to math? ② The selection criterion relies on the statistical features of code, yet the conclusion claims to have discovered an independent "cognitive scaffold" signal within the math domain. This creates a logical circularity—the causal relationship between the selection criterion and the conclusion becomes inextricably entangled.

- Experimental Design Cannot Distinguish "Capacity Competition" from "Resource Competition": The ablation experiments are conducted under a fixed total token budget. Observing that increasing the proportion of code degrades math performance is interpreted by the authors as evidence of "capacity-level competition" between domains. However, this could entirely be explained by resource-level competition: code data simply crowds out the training budget for math data, leading to under-exposure of the model to math. The experimental design lacks control groups to differentiate these two mechanisms (e.g., keeping math tokens constant while increasing total budget). Therefore, the causal interpretation that "code harms math" lacks evidentiary support.

---

> ### Author Rebuttal · Authors · 2026-03-25
>
> We greatly appreciate your valuable feedback. Our response is as follows.
> ## W1 & Q1
>
> > The reviewer questions whether the observed gain reflects any special benefit of a "code+text hybrid format" at all, rather than simply arising from reasoning-rich content that already contains explicit intermediate reasoning processes.
>
> **Response:**
>
> This concern does not apply to our main ablation. We explicitly separate pure code from Code-NL and keep Code-NL while ablating only pure code, so our result is not driven by a pure-code versus code-plus-text comparison, nor do we claim that hybrid formatting itself improves reasoning.
>
> Our point is narrower: prior work may attribute gains to "code" even when the training mix includes structured, code-adjacent data such as Markdown, HTML, or CSS, which often contains explanations and intermediate reasoning traces. Such data may help reasoning, but that does not show that pure code is the source of the gain.
>
> Thus, our conclusion is not that a hybrid format is beneficial per se, but that the useful signal likely comes from structured reasoning traces in cross-domain data. Our contribution is to disentangle this signal from pure code and clarify why the claim that "code improves reasoning" can be confounded.
>
> ## W2 & Q2
>
> > The reviewer worries that our notion of "cognitive scaffolds" may be circular: if the FastText selector is trained from code-derived positives and makes math data statistically resemble code, why would those code-like patterns help while pure code itself remains harmful?
>
> **Response:**
>
> We do not view cognitive scaffolds as circular or as simply "making math look like code." The FastText selector is only an operational tool: it captures organizational patterns common in well-structured reasoning traces, such as explicit subgoals, intermediate states, case splits, and verification steps, rather than executable program semantics.
>
> This is why scaffolded math can help while pure code can still hurt under fixed-budget training. Pure code often compresses computation and omits the explicit decomposition and justification that support mathematical learning. By contrast, the selected samples remain semantically mathematical but present the reasoning process in a clearer, more structured form.  For example, consider an olympiad-style number theory problem asking for Find all integers $n$ such that $n^2+n+1 | n^5+n^2+1$. A structured reasoning trace may proceed as follows:
>
> ````
> Step1: setup
>     Let $d = n^2 + n + 1.$, We aim to evaluate:
>     $n^5 + n^2 + 1 \pmod{d}.$
> Step2: Key Modular Reduction
>     Observe: $n^3 - 1 = (n - 1)(n^2 + n + 1)$, so $n^3 \equiv 1 \pmod{d}.$
>     Then: $n^5 = n^2 \cdot n^3 \equiv n^2 \pmod{d}.$
>     Thus: $n^5 + n^2 + 1 \equiv 2n^2 + 1 \pmod{d}.$
> Step3 - n: ... (For the sake of readability, the following steps have been omitted.)
>
> Computational Verification:
> ```python
> def check(n):
>     d = n*n + n + 1
>     val = n**5 + n**2 + 1
>     return val % d == 0
>
> solutions = []
> for n in range(-10, 11):
>     if check(n):
>         solutions.append(n)
>
> solutions
> ```
> Expected output:
> ```python
> [-2, -1, 0, 1]
> ```
> ````
>
> This kind of sample contains explicit subgoals, intermediate lemmas, and dependency structure that support mathematical reasoning, even though it contains no executable program semantics.
>
> So the beneficial factor is not "code-ness," but structured reasoning organization. Code-derived positives are simply a practical proxy for training the selector; they do not imply that the target concept is identical to code or that pure code should help math.
>
> ## W3 & Q3
>
> > The reviewer argues that, under a fixed-budget design, the observed effect may reflect simple differential data exposure or resource allocation rather than genuine capacity competition, and therefore may not causally support the claim that code harms math.
>
> **Response:**
>
> The fixed-budget setting is intentional, because competition between data sources under finite training resources is exactly the practical regime of large-scale model training. Whether framed as resource competition, capacity competition, or differential exposure, the key question is the same: when code replaces math-oriented tokens under a fixed budget, what happens to mathematical capability?
>
> Our experiments answer this directly: increasing the proportion of generic code consistently weakens downstream math performance. This is enough to challenge the common view that more code is broadly helpful for reasoning. In this setting, differential exposure is not a nuisance variable; it is the way competition manifests in practice.
>
> We therefore do not claim to isolate a single internal mechanism. Rather, we show that under realistic fixed budgets, data composition materially shapes downstream capability and different domains can compete rather than contribute independently. We further show that carefully selected cross-domain structured data can mitigate this competition.

---

> > ### Author Rebuttal · Reviewer_EDiN · 2026-04-01
> >
> > The clarifications on W1 and W2 are noted but remain largely argumentative without new empirical evidence. The response to W3 is unsatisfactory—conflating resource competition with capacity competition does not resolve the concern; it sidesteps it. A small-scale proxy experiment (e.g., 1B model with fixed math token count) is entirely feasible and should be provided to substantiate the causal claims central to this paper.

---

> > > ### Author Response · Authors · 2026-04-05
> > >
> > > Thank you for the follow-up. Your comments are important to us, and we appreciate the opportunity to clarify the remaining concerns and to provide an additional controlled experiment for W3.
> > >
> > > First, for W1 and W2, we believe the remaining disagreement mainly comes from a misunderstanding of what our paper actually varies and what it does not. In our main code ablation, we do not ablate Code-NL as an experimental variable, nor do we claim that a generic "code+text hybrid format" is itself the reason for better reasoning. Our setup instead keeps Code-NL fixed and ablates only pure code. For that reason, we are not sure how the conclusion that our claim is simply that "code-NL improves reasoning because it is high-quality reasoning data" is being attributed to us, since this is not the comparison made in our experiments. Our point is narrower: prior work may have grouped together pure code and cross-domain structured data, whereas our paper tries to disentangle these sources. The same issue applies to W2. Our goal there is to identify high-quality, high-knowledge-density mathematical subsets, using structured features as a signal. In this context, "looking code-like" refers only to external organizational structure, while "being pure code" means containing programming semantics and programming knowledge. These two should not be conflated. We also note that the comment and acknowledgement does not specifically ask for additional empirical evidence on W1 or W2. As a result, our response on these two points has mainly focused on clarifying the underlying misunderstanding and restating the actual experimental setup more precisely. We respectfully ask that these points be judged based on a faithful reading of what the paper actually studies.
> > >
> > > Second, for W3, we agree that an additional control is valuable, and we have now conducted the suggested small-scale proxy experiment on a 1B dense model. In this experiment, we keep the math token count fixed and then add an additional 50\% code tokens, so the intervention no longer comes from reducing math exposure. This directly addresses the concern that the original fixed-budget result might be explained only by differential math-token allocation. Under this setting, we observe the following results on math and code benchmarks:
> > >
> > > ### Math Benchmarks
> > >
> > > | Benchmark     | Base (fixed math tokens) | +50% code tokens | Delta   |
> > > | ------------- | :----------------------: | ---------------- | ------- |
> > > | GSM8K         |          58.00           | 54.80            | -5.52%  |
> > > | MATH          |          31.00           | 27.40            | -11.61% |
> > > | Minerva Math  |           5.15           | 2.21             | -57.09% |
> > > | OlympiadBench |           4.44           | 4.30             | -3.15%  |
> > >
> > > ### Code Benchmarks
> > >
> > > | Benchmark     | Base (fixed math tokens) | +50% code tokens | Delta   |
> > > | ------------- | ------------------------ | ---------------- | ------- |
> > > | MBPP          | 19.80                    | 20.60            | +4.04%  |
> > > | CruxEval      | 13.69                    | 14.54            | +6.21%  |
> > > | LiveCodeBench | 0.82                     | 1.63             | +98.78% |
> > > | CodeForces    | 3.93                     | 4.57             | +16.28% |
> > >
> > > This new control is intended precisely to separate the "less math exposure" explanation from the broader competition effect. If performance still changes materially when math tokens are held constant, then the concern cannot be explained purely by replacing math with code under a fixed budget.
> > >
> > > Finally, your review is highly important to us because it pushed us to sharpen both our wording and our empirical support. At the same time, we sincerely hope you can reconsider W1 and W2 in light of the clarifications above, since we believe those concerns stem from a misunderstanding of our actual experimental design rather than from a substantive flaw in the paper's evidence. With the additional W3 control experiment now included, we respectfully ask whether you would consider updating your score.

---

### Official Review · Reviewer_tdpW · 2026-03-10

**Soundness:** 3
**Presentation:** 2
**Significance:** 3
**Originality:** 2
**Overall Recommendation:** 4
**Confidence:** 3

**Summary:**

This paper studies how different data formulations affect model capabilities, especially whether the benefit of code data actually comes from more general structured reasoning signals. The experiments are extensive and the topic is interesting, but several key concepts and results still need clearer definition and explanation.

**Compliance With Llm Reviewing Policy:**

Affirmed.

**Final Justification:**

partially solved. But I still feel confused about the main contribution as the main evaluation method does not have a definition or an equation to calculate the score.

**Key Questions For Authors:**

see weakness

**Limitations:**

yes

**Strengths And Weaknesses:**

Strength:

The paper provides extensive experiments on how different data formulations affect model performance, which is interesting and potentially valuable to the community.

Weakness:

The main concepts are still too abstract. For example, the definition of “structured reasoning” is unclear. How is it identified in practice? Are there any explicit metrics or criteria to determine whether a sample belongs to this category?

To support the claim about data distribution, it would also be helpful to include mixture-training experiments across different domains or different levels of structural formulation. In particular, if the claimed structure mainly relies on special symbols or formatting patterns, the paper should provide corresponding dataset statistics.

Similarly, the definition of the five capability dimensions is not sufficiently clear. Are there concrete metrics or benchmark assignment rules for evaluating these abilities?

In addition, the first row of Figure 5 is difficult to interpret. The authors should better explain what “usage frequency” means and how readers should understand these results.

---

> ### Author Rebuttal · Authors · 2026-03-27
>
> ## W1
> > The reviewer raises a concern that the notion of structured reasoning remains insufficiently concrete and asks for a clearer operational definition and identification criterion.
>
> **Response:**
>
> We believe this concept is already defined in the main text, but we agree that the operational criterion can be stated more explicitly.
>
> In our paper, “structured reasoning” does not refer to a vague abstraction. It specifically denotes samples with explicit symbolic structure, hierarchical derivation steps, and clear cross-step logical dependencies, so that intermediate reasoning trajectories are visible and traceable.
>
> In practice, this category is identified through our cognitive scaffold construction pipeline rather than subjective manual judgment. Specifically, we train a lightweight FastText classifier to distinguish highly structured text from ordinary unstructured text. We then apply this classifier to the math corpus and treat the positively identified samples as structured reasoning data, i.e., cognitive scaffolds.There are explicit criteria and metrics. A sample belongs to this category if it is classified as positive on a validation set by maximizing F1. As reported in the appendix, this classifier achieves 0.9696 accuracy.
>
> To make this more concrete, we will also add a representative positive example of cognitive scaffolds in the revision, such as:
>
> ````
> Step1: Setup
>     Let $d = n^2 + n + 1.$, We aim to evaluate:
>     $n^5 + n^2 + 1 \pmod{d}.$
> Step2: Key Modular Reduction
>     Observe: $n^3 - 1 = (n - 1)(n^2 + n + 1)$, so $n^3 \equiv 1 \pmod{d}.$
>     Then: $n^5 = n^2 \cdot n^3 \equiv n^2 \pmod{d}.$
>     Thus: $n^5 + n^2 + 1 \equiv 2n^2 + 1 \pmod{d}.$
> ... (For the sake of readability, the following steps have been omitted.)
>
> Computational Verification:
> ```python
> def check(n):
>     d = n*n + n + 1
>     val = n**5 + n**2 + 1
>     return val % d == 0
> ```
> ````
> ## W2
>
> > The reviewer suggests that our claim about data distribution would be better supported by additional dataset-level statistics on structural formulation.
>
> **Response:**
>
> We appreciate this suggestion. We agree that dataset-level statistics are useful for making the notion of structural formulation more concrete. We compare a MATH with cognitive scaffolding data and report the statistics:
>
> | Feature | MATH| Cognitive scaffolding |
> | :----------- | :------- | :--------|
> | Symbol density |0.0363| 0.0518 |
> | Average derivation steps | 3.3720| 4.4124|
> | Indentation ratio| 0.0006   | 0.5446|
> | Text length|531| 2821|
>
> These statistics suggest that the structural properties of our cognitive scaffolding data are reflected in longer derivational organization, substantially higher indentation-based structure, and greater overall compositional length.
>
> ## W3
>
> > The reviewer points out that the definitions of the five capability dimensions are not sufficiently clear and asks for explicit benchmark assignment rules and evaluation criteria.
>
> **Response:**
>
> The five capability are defined through an explicit benchmark-to-dimension mapping used consistently in our evaluation pipeline. Concretely, each benchmark is assigned to one primary capability dimension according to its evaluation target. As shown in the table below:
>
> | Capability dimension|Benchmarks|
> | :------------ | :--|
> | General knowledge| ARC-c, ARC-e|
> | Program ability| HumanEval, HumanEval+, MBPP, MBPP+, CruxEval, BIRD-SQL, CodeForces, LiveCodeBench, BigCodeBench |
> | Mathematical ability| GSM8K, GSM-Plus, CMath, MATH, Minerva-Math, OlympiadBench, CollegeMath, MathBench|
> | Comprehensive reasoning| PIQA, HellaSwag|
> | Professional knowledge|MMLU, GPQA|
>
> Accordingly, each capability dimension is computed as an aggregate score over the benchmarks assigned to that dimension.
>
> ## W4
>
> > The reviewer notes that the first row of Figure 5 is difficult to interpret and requests a clearer explanation of the meaning of “usage frequency” and how the figure should be read.
>
> **Response:**
>
> “Usage frequency” refers to the token-level expert usage count in the MoE model. Concretely, during autoregressive generation, each token is routed to a subset of experts, and we count how many times each of the 64 experts is selected over all tokens. Therefore, the first row of Figure 5 shows which experts are used more often by tokens from that domain. This plot should be interpreted as a visualization of the model’s internal expert allocation pattern: if several experts have much higher counts, the model relies more heavily on those experts for that domain. The main takeaway is that removing a domain-specific corpus changes the corresponding domain’s expert usage pattern most clearly. The configuration involving cognitive scaffolds remains much closer to the original expert usage pattern, suggesting that such data improve reasoning without substantially disturbing the model’s internal expert organization.

---

> > ### Author Rebuttal · Reviewer_tdpW · 2026-04-01
> >
> > I still don't know what cognitive scaffolding is, how you define it, or how to calculate it. For the first line of Figure 5, I still feel the figures show random lines, and I can't distinguish the differences between these figures. Will keep the original score

---

> > > ### Author Response · Authors · 2026-04-05
> > >
> > > Thank you for the follow-up. We respectfully think there may still be a misunderstanding about what we mean by cognitive scaffolding. In our paper, cognitive scaffolding is not introduced as a fully formalized mathematical object, but rather as a high-quality, knowledge-dense subset of mathematical data that provides stronger intermediate support for reasoning. While we do not claim a universal formal definition, both the main paper and our rebuttal already explain how this subset is constructed in operational terms. In other words, the notion is meant to be understood through its concrete construction procedure in our setting, rather than through an abstract standalone axiomization.
> > >
> > > For Figure 5, the first row shows the distribution of expert routing. We agree that these raw routing patterns may be difficult to distinguish reliably by visual inspection alone. This is precisely why the second row is included: it provides a more fine-grained statistical view of the differences in these distributions, which is the intended evidence supporting our interpretation. So the first row is mainly a qualitative illustration, while the second row contains the more informative quantitative comparison.
> > >
> > > More broadly, we would like to emphasize that these presentation-level concerns are not fully aligned with the core contribution of the paper. The main contribution is the identification and empirical validation of the role of cognitively scaffolded, high-quality mathematical data in improving reasoning behavior. Whether a term is fully formalized, or whether a qualitative visualization is immediately visually salient in its raw form, does not materially change the central empirical findings and conclusions, which are supported by the full set of quantitative results.
> > >
> > > Since your score is important for the final outcome, we sincerely hope you might reconsider whether the remaining concerns here warrant keeping the original score unchanged, especially given that they relate more to presentation clarity than to the validity of the core claims.

---

### Official Review · Reviewer_bYs5 · 2026-03-13

**Soundness:** 3
**Presentation:** 4
**Significance:** 3
**Originality:** 3
**Overall Recommendation:** 5
**Confidence:** 4

**Summary:**

**Contribution**
This work investigates how code and math corpora affect a model’s capabilities across several domains: general knowledge, programming ability, mathematical reasoning, comprehensive reasoning, and professional knowledge. Through ablation studies and mechanistic interpretability analyses, the authors reveal a competitive effect of code corpora on knowledge-intensive tasks, particularly complex mathematical problems, while mathematical corpora exhibit competition with comprehensive reasoning tasks. The work further identifies a subset of functions that serve as “cognitive scaffolding” for mathematical reasoning, particularly in complex problem-solving scenarios. Based on these findings, the authors propose a promising data-centric optimization strategy: incorporating cross-domain synergistic data (i.e., cognitive scaffolding) can preserve performance in one domain while enhancing performance in another.

**Implementation**
The work employs Mixture-of-Experts (MoE) models by replacing the standard feed-forward network (FFN) with a collection of \(N\) experts, each implemented as a compact modular FFN unit. With the total number of model parameters held constant, the authors increase the number of experts while proportionally reducing the intermediate dimensionality of each expert. This configuration encourages stronger specialization among experts.

For data curation, the authors collect data from multiple public repositories and ensure high quality through a strict admission process. Data from each domain are combined using a dynamic sampling policy that balances token-level contributions while prioritizing high-value domains such as mathematics and code. The authors then conduct ablation studies by removing either the pure code corpus or the mathematical corpus. Corpora from the remaining domains are proportionally upsampled to compensate for the removed data.

**Compliance With Llm Reviewing Policy:**

Affirmed.

**Key Questions For Authors:**

Some key information remains unclear:

1. After the cognitive scaffolds are identified, how are they incorporated into the training data? What percentage of the original data is replaced with cognitive scaffolds? Does a higher replacement percentage yield stronger reasoning capability? What sampling strategy is applied to the cognitive scaffolding data?

2. Is the cross-domain learning capability introduced by cognitive scaffolding bidirectional? Regarding how the cognitive scaffolding data is incorporated, the work conducts a one-directional study: replacing mathematical corpora with cognitive scaffolding data. The paper could benefit from the reverse experiment—replacing code corpora with cognitive scaffolding data. The absence of this study weakens the claim regarding cross-domain learning enabled by cognitive scaffolding.

**Limitations:**

Yes

**Strengths And Weaknesses:**

## Soundness: Good ##

The work conducts comprehensive ablation studies across various configurations and benchmarks. The methodology of leveraging MoE activation patterns is solid, and important implementation details are well addressed, including total token budgets, shared experts, the dropless routing strategy, load-balancing loss, and router z-loss. The clean data curation pipeline further strengthens the study’s soundness—for example, the clear separation between Code and Code-NL and the strict admission logic. The differences from prior studies are well explained and addressed. The competitive relationship between Code data and Math data is carefully explored and clearly illustrated (Figures 2 and 3).

Cognitive scaffolding is the key finding of the work. Proper construction of the data—for example, using FastText—provides a solid foundation for the subsequent analysis. Extensive ablation studies strengthen the argument that cognitive scaffolding enhances complex reasoning while preserving stability in expert activation distributions.

The extension to dense models is the icing on the cake. This experiment greatly strengthens the generality of the findings and the usability of the methodology. In particular, it eliminates the concern that the observed specialization is caused by router-policy artifacts and further supports the claim that the effect originates at the data level.

The trade-offs are clearly and honestly presented. For example, on relatively simple tasks (e.g., GSM8K and CMath), structured reasoning can disrupt cases that could otherwise be solved directly via natural language, leading to performance drops of 6.29% and 2.00%.

Some key information remains unclear:

1. After the cognitive scaffolds are identified, how are they incorporated into the training data? What percentage of the original data is replaced with cognitive scaffolds? Does a higher replacement percentage yield stronger reasoning capability? What sampling strategy is applied to the cognitive scaffolding data?

2. Is the cross-domain learning capability introduced by cognitive scaffolding bidirectional? Regarding how the cognitive scaffolding data is incorporated, the work conducts a one-directional study: replacing mathematical corpora with cognitive scaffolding data. The paper could benefit from the reverse experiment—replacing code corpora with cognitive scaffolding data. The absence of this study weakens the claim regarding cross-domain learning enabled by cognitive scaffolding.

## Presentation: Excellent ##

The narrative is logically coherent, smooth, and easy to follow. Figures are descriptive and convey information effectively. Details are clearly presented and well explained. The paper is pleasant to read.

## Significance: Good ##

The work revisits the prior claim that “code corpus enhances reasoning ability” and provides a counterargument supported by deeper analysis. It further identifies cognitive scaffolding as a mechanism that enhances cross-domain interaction while preserving the stability of expert distributions. The work illuminates a promising direction for data-centric model optimization: by incorporating cognitive scaffolding, training can preserve performance in one domain while improving performance in another.

## Originality: Good ##

The most original element of the work is the Code/Math bidirectional ablation. To the best of my knowledge, prior work has not conducted a clear and systematic code–math ablation study.

The work demonstrates additional novelty by combining three components:
1. MoE expert activation as an analysis signal
2. strict separation between Code and Code-NL
3. Code/Math bidirectional ablation (to the best of my knowledge, prior work has not conducted a clear code–math ablation study)

The work provides numerous salient ablation studies, stronger quantitative analysis, and derives novel conclusions that prior work did not reach.

---

> ### Author Rebuttal · Authors · 2026-03-28
>
> ## Q1
>
> > The reviewer asks for a clearer description of how cognitive scaffolds are incorporated into training, including what is replaced, the replacement proportion, whether higher replacement ratios help more, and what sampling strategy is used.
>
> **Response:**
>
> We thank the reviewer for this helpful question and agree that the integration procedure should be described more explicitly.
>
> In our current design, cognitive scaffolds are introduced within the existing math-domain budget, rather than by adding extra tokens or increasing the overall proportion of math data. Concretely, after identifying structured reasoning samples from the math corpus using the FastText selector, we replace part of the original math data with these scaffold samples while keeping the total math-token budget unchanged. This is the setting referred to in the paper as "holding the overall mathematical token budget constant" and "under the same budget for math-domain data."
>
> Therefore, the main point of the experiment is not that more math data helps, but that a better internal composition of math data helps. The gain comes from replacing part of ordinary math data with more structured reasoning data, rather than from increasing total exposure.
>
> Regarding sampling, the scaffolded data are incorporated through the same training pipeline used in our other settings. They are not treated as an additional domain with a special schedule; instead, after replacement within the math budget, they are sampled with the same uniform sampling strategy as the other settings.
>
> At the same time, we would like to be precise about scope: the current paper does not present a systematic sweep over multiple replacement ratios, so we do not claim that "more replacement is always better." What our current evidence supports is a narrower result: under a fixed math budget, replacing part of the original math data with structured reasoning data substantially improves difficult mathematical reasoning benchmarks, while introducing the expected trade-off on simpler tasks such as GSM8K and CMath. We will revise the paper to make both the implementation detail and this scope limitation explicit.
>
> ## Q2
>
> > The reviewer asks whether the cross-domain benefit of cognitive scaffolding is truly bidirectional, and suggests that the paper would be stronger if it also included the reverse experiment of replacing code corpora with cognitive scaffolding data.
>
> **Response:**
>
> We thank the reviewer for raising this important point. While we understand the concern about whether the cross-domain benefit should be validated in a fully bidirectional manner, we would like to clarify that cognitive scaffolds are not an external data source parallel to both Math and Code. Rather, they are a structured subset of the math corpus itself.
>
> This distinction matters for interpreting the intervention. In our current setting, replacing part of the math corpus with cognitive scaffolds increases the density of high-quality structured reasoning signals within the math budget, which is why it improves difficult mathematical reasoning. By contrast, using the same scaffold data to replace code corpus would not be a symmetric reverse test: it would effectively substitute programming data with math-derived structured data.
>
> Under that intervention, our expectation is that programming performance would drop sharply, because genuine code data are being removed, while mathematical reasoning might improve only modestly due to the higher concentration of structured reasoning signals.

---

### Official Review · Reviewer_rXbC · 2026-03-13

**Soundness:** 4
**Presentation:** 3
**Significance:** 3
**Originality:** 3
**Overall Recommendation:** 4
**Confidence:** 3

**Summary:**

The paper revisits and challenges the prevailing view that incorporating code data generally enhances the reasoning abilities of large language models. Through large-scale controlled experiments on MoE models with different expert configurations and on 1B/5B dense models, the study shows that pure code data creates significant resource competition and negative interference with knowledge-intensive tasks such as mathematics. On this basis, the authors propose and validate the true key to improving complex reasoning capabilities: “Cognitive Scaffolds” data with a rigorous logical structure.

**Compliance With Llm Reviewing Policy:**

Affirmed.

**Final Justification:**

The authors' rebuttal effectively addressed my concerns regarding the Cognitive Scaffolds extraction, performance trade-offs, and data purity. Given the paper's rigorous experiments and valuable mechanistic insights into MoE models, it remains a technically solid contribution. Therefore, I maintain my score of Weak Accept (4).

**Key Questions For Authors:**

- How pure is the pure code data? Could the remaining natural language components still affect the model’s reasoning ability?
- How can the performance drop of Cognitive Scaffolds on simple tasks be mitigated? In practical training, how can one balance simple natural language reasoning and complex logical reasoning at the same time?

**Limitations:**

The paper’s method for extracting Cognitive Scaffolds is relatively shallow, as it relies solely on a FastText classifier. Moreover, while it improves performance on complex mathematical reasoning, it inevitably comes at the expense of the model’s performance on simpler natural language reasoning tasks, such as GSM8K. In addition, the paper does not provide a detailed analysis of whether the “negative coupling” effect between code and mathematics is actually caused by limitations in model capacity.

**Strengths And Weaknesses:**

Strengths:

- The experiments are rigorous and conducted at a relatively large scale. They cover MoE models with different numbers of experts as well as Dense models at the 1B and 5B scales, and include large-scale controlled comparisons across multiple architectures and model sizes. The study also ensures the rigor of the experimental design through strict variable control, such as fixing the number of training tokens and using proportionally compensated ablation data.
- The paper extends its analysis to the internal mechanisms of the models, conducting an in-depth investigation of expert routing distributions in MoE models. Rather than relying solely on benchmark scores to demonstrate effectiveness, it also provides mechanistic insights from within the model.

Weaknesses:

- The mechanism for extracting Cognitive Scaffolds is relatively simple, relying only on a lightweight FastText classifier. Such a classifier is quite shallow and can only capture surface-level formatting features, rather than extracting scaffolds based on deeper logical structure.
- Cognitive Scaffolds data appears to come at the cost of performance on simpler reasoning tasks, meaning this type of data is not Pareto-optimal.
- One of the paper’s core findings is that there exists a significant “negative coupling” effect between data from different domains, such as pure code and mathematics. However, this phenomenon was observed only in Dense models up to 5B parameters and in MoE models of corresponding scale. In small- and medium-scale models, parameter capacity itself is already under a strict bottleneck, so knowledge from different domains is naturally prone to crowding each other out as they compete for limited representational space. It remains unclear whether this issue would still persist in larger-capacity models.

---

> ### Author Rebuttal · Authors · 2026-03-28
>
> ## W1
>
> > The reviewer is concerned that our Cognitive Scaffold extraction is too shallow because it relies on a lightweight FastText classifier, which may capture only surface formatting cues rather than deeper logical structure.
>
> **Response:**
>
> We agree that our classifier is lightweight, but lightweight does not mean ineffective. The key issue is not model size itself, but whether the selected data reflect a meaningful reasoning signal rather than a trivial formatting artifact.
>
> To clarify this, we compare our Cognitive Scaffolding data with the original MATH corpus using several structural statistics:
> | Feature| MATH|Cognitive scaffolding|
> | :-- | :--| :-- |
> | Symbol density | 0.0363 | 0.0518|
> | Average derivation steps | 3.3720 | 4.4124|
> | Indentation ratio| 0.0006 | 0.5446 |
> | Text length| 531| 2821|
>
> Although the classifier does not explicitly learn semantics, the selected samples are not merely distinguished by superficial visual patterns. They show longer derivational organization, much stronger indentation-based structure, and greater overall compositional length, all of which are consistent with more explicit reasoning trajectories.
>
> This is the motivation behind our “less is more” design: a lightweight and scalable filter can still recover a useful subset with richer structural organization, which is especially valuable in industrial-scale pretraining where extraction cost matters greatly.
> ## W2 & Q2
>
> > The reviewer notes that Cognitive Scaffolds improve complex mathematical reasoning but reduce performance on simpler reasoning tasks, and asks how this trade-off can be mitigated in practice.
>
> **Response:**
>
> We agree that Cognitive Scaffolds are not Pareto-optimal across all reasoning tasks. The drop on simpler tasks such as GSM8K is relatively modest, around 2.5%, rather than a severe degradation of basic reasoning ability.
>
> We believe this trade-off arises because Cognitive Scaffold filtering preferentially retains data with stronger explicit structure while filtering out some less-structured NL reasoning data. Thus, the training mixture becomes more specialized toward complex derivational reasoning, which helps difficult mathematical tasks but is slightly less aligned with the simpler reasoning style often sufficient for tasks like GSM8K.
>
> In practice, we view this as a manageable trade-off rather than a fundamental limitation. Simpler NL reasoning is typically easier to recover, since it relies more on broadly available supervision and less on rare high-quality structural signals. A practical strategy is therefore to use Cognitive Scaffolds to strengthen complex reasoning while retaining or reintroducing a moderate amount of unfiltered reasoning data to preserve simpler reasoning performance.
> ## W3
> > The reviewer questions whether the observed negative coupling between code and mathematics is mainly a consequence of limited model capacity at the tested scales, and whether the same effect would persist in larger models.
>
> **Response:**
>
> We agree that extrapolation to much larger models should be made cautiously. We study this phenomenon in realistic pretraining setups across multiple model families, including MoE models as well as 1B and 5B Dense models, and observe a qualitatively similar pattern: increasing the proportion of generic pure code under a fixed training budget is consistently associated with weaker downstream mathematical performance.
>
> We therefore view this result as a repeated empirical regularity within the tested pretraining regime, rather than an artifact of one isolated model scale. Our claim is not that negative coupling must hold identically in all larger-capacity models, but that it already appears across several substantially different settings and should be taken seriously in pretraining mixture design.
> ## Q1
> > The reviewer asks how "pure" the pure-code data really is, and whether residual natural-language content might still affect the observed reasoning behavior.
>
> **Response:**
>
> Our main ablation is designed precisely to separate pure code from mixed code-text data. In our corpus construction, pure code is mainly collected from open-source GitHub repositories and is defined as executable functions or program segments intended to solve programming-related problems. During curation, code files are filtered by syntax validity, length, duplication, and functional density; in domain categorization, only samples with code density above 60% and clear programming intent are assigned to the Code domain.
>
> We explicitly exclude explanatory text, comments, and instructional descriptions from what we call pure code, while keeping Code-NL in the corpus and ablating only pure code. This means the main result is not driven by removing mixed reasoning data that happens to contain code. Instead, we are specifically testing the contribution of standalone algorithmic code after separating it from cross-domain code-text mixtures.

---

> > ### Author Rebuttal · Reviewer_rXbC · 2026-04-04
> >
> > Thank you for the convincing rebuttal that clears up my concerns; I maintain my current score.

---

### Decision · Program_Chairs · 2026-04-30

**Decision:**

Accept (regular)

**Comment:**

This paper challenges the assumption that code data broadly improves LLM reasoning, showing through large-scale ablations across MoE and dense models that pure code competes with mathematical reasoning under fixed training budgets. The identification of "cognitive scaffolds", structured reasoning subsets that improve complex math reasoning, is practical and actionable. Reviewers acknowledged the experimental rigor, clean code/Code-NL separation, and MoE routing analysis. The key concern about confounding resource and capacity competition was resolved by a controlled experiment holding math tokens fixed while adding code, which still showed math degradation.